# In search of projectively equivariant networks

**Georg Bökman**[*]                                               *bokman@chalmers.se*
*Department of Electrical Engineering*
*Chalmers University of Technology*

**Axel Flinth**[*]                                               *axel.flinth@umu.se*
*Department of Mathematics and Mathematical Statistics*
*Umeå University*

**Fredrik Kahl**                                               *fredrik.kahl@chalmers.se*
*Department of Electrical Engineering*
*Chalmers University of Technology*

**Reviewed on OpenReview:** *https://openreview.net/forum?id=Ls1E16bTj8*

## Abstract

Equivariance of linear neural network layers is well studied. In this work, we relax the equivariance condition to only be true in a projective sense. Hereby, we introduce the topic of projective equivariance to the machine learning audience. We theoretically study the relation of projectively and linearly equivariant linear layers. We find that in some important cases, surprisingly, the two types of layers coincide. We also propose a way to construct a projectively equivariant neural network, which boils down to building a standard equivariant network where the linear group representations acting on each intermediate feature space are lifts of projective group representations. Projective equivariance is showcased in two simple experiments. Code for the experiments is provided at github.com/usinedepain/projectively_equivariant_deep_nets

## 1 Introduction

Deep neural networks have been successfully applied across a large number of areas, including but not limited to computer vision (Krizhevsky et al., 2012), natural language processing (Devlin et al., 2019), game play (Silver et al., 2018) and biology (Jumper et al., 2021). In many of these areas, the data has geometric properties or contains symmetries that can be exploited when designing neural networks. For instance, AlphaFold (Jumper et al., 2021) models proteins in 3D-space while respecting translational and rotational symmetries. Much of the work on neural networks respecting geometry and symmetry of the data can be boiled down to formulating the symmetries in terms of *group equivariance*. Group equivariance of neural networks is a currently very active area of research starting with (Wood & Shawe-Taylor, 1996) and brought into the deep networks era by Cohen & Welling (2016). Group equivariant neural networks are part of the broader framework of *geometric deep learning*. Recent surveys include (Bronstein et al., 2021; Gerken et al., 2021). Our interests in equivariance are driven by applications in computer vision, (Bökman et al., 2022; Bökman & Kahl, 2022; 2023), but in this paper, we will give application examples in several different research fields, targeting a more general machine learning audience.

Given a set of transformations $\mathcal{T}$ acting on two sets $X$, $Y$, a function $f : X \to Y$ is called *equivariant* if applying $f$ commutes with the transformations $t \in \mathcal{T}$, i.e.,

$$t[f(x)] = f(t[x]). \tag{1}$$

---

[*]Equal contribution

Concretely, consider rotations acting on point clouds and $f$ mapping point clouds to point clouds. If applying $f$ and then rotating the output yields the same thing as first rotating the input and then applying $f$, we call $f$ rotation equivariant. In the most general scenario, $t \in \mathcal{T}$ could act differently on $X$ and $Y$; For instance, if the action on $Y$ is trivial, we can model invariance of $f$ in this way.

If the transformations $\mathcal{T}$ form a group, results from abstract algebra can be used to design equivariant networks. A typical way of formulating an equivariance condition in deep learning is to consider the neural network as a mapping from a vector space $V$ to another vector space $W$ and requiring application of the network to commute with group actions on $V$ and $W$. As $V, W$ are vector spaces the setting is nicely framed in terms of representation theory (we will provide a brief introduction in Section 2). What happens however when $V$ and/or $W$ are not vector spaces? This work covers the case when $V$ and $W$ are projective spaces, i.e., vector spaces modulo multiplication by scalars. This generalization is motivated, for instance, by computer vision applications. In computer vision it is common to work with homogeneous coordinates of 2D points, which are elements of the projective space $\mathrm{P}(\mathbb{R}^3)$. In Section 2, we will describe more examples.

In the vector space case, any multilayer perceptron can be made equivariant by choosing the linear layers as well as non-linearities equivariant. This is one of the core principles of geometric deep learning. In this paper, we will theoretically investigate the consequences of applying the same strategy for building a projectively equivariant (to be defined below) multilayer perceptron, i.e. one that fulfills (1) only up to a scalar (which may depend on $x$). More concretely, we will completely describe the sets of projectively invariant linear layers in a very general setting (Theorem 2.15).

The most interesting consequence of Theorem 2.15 is a negative one: In two important cases, including the SO(3)-action in the pinhole camera model and permutations acting on tensors of not extremely high order, the linear layers that are equivariant in the standard sense (*linearly equivariant*) are *exactly the same* as the projectively equivariant ones. This is surprising, since projective equivariance is a weaker condition and hence should allow more expressive architectures. Consequently, any projectively equivariant multilayer perceptron must in these cases either also be linearly equivariant, or employ non-trivial projectively equivariant non-linearities.

Theorem 2.15 also suggests a natural way of constructing a projectively equivariant network that for certain groups is different from the linearly equivariant one. These networks are introduced in Section 3. We then describe a potential application for them: Classification problems with class-varying symmetries. In Section 3.2, we describe this application, and perform some proof-of-concept experiments on modified MNIST data. In Section 4 we generalize Tensor Field Networks (Thomas et al., 2018) to spinor valued data, whereby we obtain a network that is equivariant to projective actions of SO(3).

It should be stressed that the main purpose of this article is *not* to construct new architectures that can beat the state of the art on concrete learning tasks, but rather to theoretically invest what projectively equivariant networks can (and more importantly, cannot) be constructed using the geometric deep learning framework. The boundaries that we establish will guide any practitioner trying to exploit projective equivariance.

We have for convenience collected important notation used in this article in Table 1. Appendix A also contains definitions of the mathematical objects we use in the article.

## 2 Projective equivariance

In this work, we are concerned with networks that are *projectively equivariant*. Towards giving a formal definition, let us first discuss how equivariance can be formulated using the notion of a *representation* of a group.

A *representation* $\widehat{\rho}$ of a group $G$ on a vector space $V$ over a field $\mathbb{F}$ is a map which associates each group element $g$ with an isomorphism $\widehat{\rho}(g) : V \to V$ in a manner that respects the structure of the group, i.e.,

$$\widehat{\rho}(g)\widehat{\rho}(h) = \widehat{\rho}(gh) \text{ for all } g, h \in G. \tag{2}$$

A more compact way of stating this is that $\widehat{\rho}$ is a *group homomorphism* from the group $G$ to the group of isomorphisms (*general linear group*) $\mathrm{GL}(V)$ on $V$.

| | | | |
|---|---|---|---|
| $P(V)$ | Projective space of $V$, i.e., $V/(\mathbb{F} \setminus \{0\})$ | $\Pi_V$ | Projection map from $V$ to $P(V)$ |
| $\mathrm{Hom}(V, W)$ | Space of linear maps from $V$ to $W$ | $\mathrm{GL}(V)$ | General linear group of $V$ |
| $\mathrm{PGL}(V)$ | Projective general linear group of $V$ | $\varphi$ | Group covering map |
| $\rho$ | Projective representation | $\widehat{\rho}$ | Linear representation |
| $[n]$ | $\{0, 1, \ldots, n-1\}$ | $\mathbb{Z}_n$ | Additive group of integers modulo $n$ |
| $S_n$ | Permutation group of $n$ elements | $A_n$ | Permutations of signature 1 |
| $\mathrm{SO}(3)$ | Rotation group in 3D | $\mathrm{SU}(2)$ | Unitary matrices in $\mathbb{C}^{2,2}$ with determinant 1 |
| id | The identity matrix | sgn | The signum function on $S_n$. |

Table 1: Symbol glossary

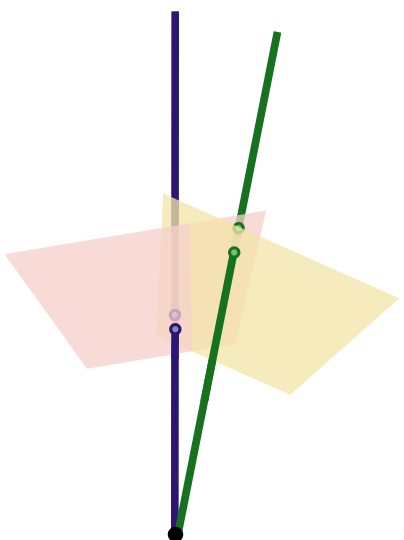

Figure 1: The pinhole camera model. The black dot is the center of projection, i.e. the camera center. The red and yellow planes represent two different image planes corresponding to two different viewing directions of a camera at the black dot. 3D points are projected to the image planes via their viewing rays — the straight lines connecting them to the center of projection. 3D rotations act on points in (say) the yellow plane in a projective manner — when we rotate the camera from the yellow plane to the red plane, points on the yellow plane do usually not lie on the red plane but can be identified with points on the red plane by projection in the camera center. E.g. the green point on the yellow plane transforms to a point on the red plane under rotation of the camera by identifying the point with the green line and then projecting this line to the red plane. When working with homogeneous coordinates in computer vision, we identify points in the image plane with their viewing rays. Rotations act on homogeneous coordinates through a projective representation. See also Example 2.1 and Example 2.5.

We also assume that $G$ is equipped with a topology, with respect to which the multiplication and the inverse operation are continuous (that is, $G$ is a *topological group*). The representations are assumed to be continuous with respect to that topology. As we will also consider *projective representations* in this paper, we will refer to 'ordinary' representations (2) as *linear representations*. Given linear representations $\widehat{\rho}_0$ and $\widehat{\rho}_1$ on spaces $V$ and $W$, respectively, we say that a map $\Phi : V \to W$ is *equivariant* with respect to $\widehat{\rho}_0$ and $\widehat{\rho}_1$ if

$$\widehat{\rho}_1(g)\Phi(v) = \Phi(\widehat{\rho}_0(g)v) \text{ for all } g \in G, v \in V. \tag{3}$$

If in (3), $\widehat{\rho}_1(g)$ is the identity transformation for every $g \in G$, we say that $\widehat{\rho}_1$ acts trivially and then $\Phi$ is called *invariant*. Many symmetry conditions can be phrased in terms of (3), and ways to design neural networks $\Phi$ satisfying (3) have been extensively studied for e.g. rigid motions in 2D/3D/$n$D (Cohen & Welling, 2016; Weiler & Cesa, 2019; Bökman et al., 2022; Bekkers et al., 2018; Thomas et al., 2018; Cesa et al., 2021), permutations of graph nodes (Maron et al., 2019; Zaheer et al., 2017; Qi et al., 2017) and for more general groups $G$ (Wood & Shawe-Taylor, 1996; Kondor & Trivedi, 2018; Cohen et al., 2020; Aronsson, 2022; Finzi et al., 2021). The most famous example is probably the translation equivariance of CNNs (Fukushima & Miyake, 1982; LeCun et al., 1998).

In this work, we want to study networks that are *projectively in- and equivariant*. To explain this notion, let us first introduce the notation $P(V)$ for the projective space associated to a vector space $V$ over a field $\mathbb{F}$. That is, $P(V)$ is the space of equivalence classes of $V \setminus \{0\}$ under the equivalence relation $v \sim w \iff \exists \lambda \in \mathbb{F} \setminus \{0\}$ s.t. $v = \lambda w$. Furthermore, we will write $\Pi_V$ for the projection that maps $v \in V$ to its equivalence class $\Pi_V(v) \in P(V)$.

*Example* 2.1. The *pinhole camera model* is well-known and popular in computer vision (Hartley & Zisserman, 2003). We illustrate it in Figure 1. In essence, the idea is to identify a point $y \in \mathbb{R}^2$ in an image with the line in $\mathbb{R}^3$ that projects to $y$ through the pinhole camera. Mathematically, this amounts to embedding $y \in \mathbb{R}^2$

to the 3D point $x = [y, 1] \in \mathbb{R}^3$, and then considering it as a point in $\mathrm{P}(\mathbb{R}^3)$. A 2D point cloud can in the same way be identified with a set of points in homogeneous coordinates, say $x_i = [y_i, 1] \in \mathbb{R}^3, i \in [m]$. In this manner, we can think of point-clouds $X$ as elements of $V = (\mathbb{R}^3)^m$. Of course, multiplying all points in $X$ with a scalar $\lambda$ does not change the image of the points, we can regard $X$ as only being defined modulo $\mathbb{R}$, i.e., as an element of $\mathrm{P}(V)$.

*Example* 2.2. In quantum mechanics, the state of a particle is determined via a $\mathbb{C}$-valued *wave-function*, that is an element $\psi$ of some Hilbert space $\mathcal{H}$, with $\langle \psi, \psi \rangle = 1$ (Hall, 2013). Any measurable property $A$ (modeled through self-adjoint operators on $\mathcal{H}$) of the particle is however determined through sesquilinear products (*brackets*) of the form $\langle \psi | A \psi \rangle$ which is not changed when $\psi$ is multiplied with a normalized scalar $\theta \in \mathbb{C}$. Physical observations can hence only reveal $\psi$ modulo such a scalar, and it should therefore be interpreted as an element of the projective space $\mathrm{P}(\mathcal{H})$.

The benefit of projective equivariance is eminent when trying to infer a projective feature transforming in a certain manner from projective data — think for example of inferring the center of mass of a point cloud (which is equivariant to rigid motions) from a photo of the cloud. Motivated by these examples, we are interested in maps that fulfil the equivariance relation (3) up to a multiplicative scalar. To formalise this notion, let us begin by introducing some notation. $\mathrm{PGL}(V)$ is the *projective general linear group*, i.e., the set of equivalence classes of maps $A \in \mathrm{GL}(V)$ modulo a multiplicative scalar. Note that an element $M \in \mathrm{PGL}(V)$ defines a projective-linear map $\mathrm{P}(V) \to \mathrm{P}(V)$.

**Definition 2.3.** A *projective representation* of a group $G$ on a projective space $\mathrm{P}(V)$ is a group homomorphism $\rho : G \to \mathrm{PGL}(V)$.

Put more concretely, a projective representation is a map associating each $g \in G$ to an equivalence class $\rho(g)$ of invertible linear maps. $\rho$ respects the group structure in the sense of (2) — however, only in the sense of elements in $\mathrm{PGL}(V)$, i.e., up to a multiplicative scalar.

*Example* 2.4. The simplest form of a projective representation is a projected linear one. That is, given a linear representation $\widehat{\rho} : G \to \mathrm{GL}(V)$, we immediately obtain a projective representation through $\rho = \Pi_{\mathrm{GL}(V)} \circ \widehat{\rho}$.

*Example* 2.5. $\mathrm{SO}(3)$ acts linearly on points in $\mathbb{R}^3$ by multiplication with the ordinary $3 \times 3$ rotation matrices $R$. The projection of this representation gives a projective representation on $\mathrm{P}(\mathbb{R}^3)$. This projective representation acts on pinhole projected points $x \in \mathrm{P}(\mathbb{R}^3)$ by multiplying any vector $v \in \mathbb{R}^3$ in the equivalence class $x$ by the ordinary $3 \times 3$ matrices $R$ and interpreting the result as lying in $\mathrm{P}(\mathbb{R}^3)$.

Not all projective representations are projections of linear ones, as shown by the next example.

*Example* 2.6. (Hall, 2013) For every $\ell = 0, 1/2, 1, 3/2, \ldots$, there is a complex vector space $\mathcal{V}_\ell$ of dimension $2\ell + 1$ on which the group $\mathrm{SU}(2)$ is acting by a linear representation. Together with the fact that $\mathrm{SU}(2)$ is a global double cover of $\mathrm{SO}(3)$, this can be used to define a map $\rho$ from $\mathrm{SO}(3)$ to $\mathrm{GL}(\mathcal{V}_\ell^n)$ up to a scalar, i.e., a projective representation. For integer $\ell$, this is a projection of a linear representation of $\mathrm{SO}(3)$, but for half-integers $1/2, 3/2, \ldots$, it is not. The parameter $\ell$ is known as the 'spin' of a particle in quantum mechanics.

Given projective representations $\rho_0$, $\rho_1$ on $\mathrm{P}(V)$ and $\mathrm{P}(W)$, respectively, we can now state equivariance of a map $\Phi : \mathrm{P}(V) \to \mathrm{P}(W)$ exactly as before in (3):

$$\rho_1(g)\Phi(v) = \Phi(\rho_0(g)v) \text{ for all } g \in G, \, v \in P(V). \tag{4}$$

Note that we only demand that the equality holds in $\mathrm{P}(W)$ and *not* necessarily in $W$, which was the case in (3). We refer to $\Phi$ satisfying (4) as being *projectively equivariant* with respect to $\rho_0$ and $\rho_1$.

## 2.1 Projectively equivariant linear maps

A canonical way to construct (linearly) equivariant neural networks is to alternate equivariant activation functions and equivariant linear layers. More concretely, if we let $V_0$ denote the input-space, $V_i$, $i = 1, \ldots, K-1$ the intermediate spaces and $V_K$ the output space, a multilayer perceptron is a function of the form

$$\Phi(v) = v_K(v), \quad v_{k+1} = \sigma_k(A_k v_k), k = 0, \ldots, K - 1.$$

It is now clear that if $A_k$ and $\sigma_k$ all are equivariant, $\Phi$ will also be. This method easily generalizes to projective equivariance: one simply needs to restrict the equivariance condition to a projective equivariance one. In the following, we will refer to such nets as *canonical*.

This construction motivates the question: Which linear maps $A : V \to W$ define projectively equivariant transformations $\mathrm{P}(V) \to \mathrm{P}(W)$? If the projective representations $\rho_0$ and $\rho_1$ are projections of linear representations $\widehat{\rho}_0$ and $\widehat{\rho}_1$ respectively, we immediately see that every equivariant $A$ defines a projectively equivariant transformation. These are however not the only ones—an $A$ satisfying

$$A\widehat{\rho}_0(g) = \varepsilon(g)\widehat{\rho}_1(g)A, \quad g \in G \tag{5}$$

for some function $\varepsilon : G \to \mathbb{F}$ will also suffice. For such a relation to hold for non-zero $A$, $\varepsilon$ needs to be a (continuous) *group homomorphism*. The set of these $\varepsilon$, equipped with pointwise multiplication, form the so-called *character group*.

**Definition 2.7.** Let $G$ be a topological group. The set of continuous group homomorphisms $\varepsilon : G \to \mathbb{F}$ is called the *character group* of $G$, and is denoted $G^*$.

*Remark* 2.8. Given an $\varepsilon \in G^*$ and a linear representation $\widehat{\rho}$ of $G$, we may define a new representation $\widehat{\rho}^{\varepsilon}$ through $\widehat{\rho}^{\varepsilon}(g) = \varepsilon(g)\widehat{\rho}(g)$. Hence, (5) means that $A$ is linearly equivariant w.r.t. representations $\widehat{\rho}_0$ and $\widehat{\rho}_1^{\epsilon}$.

The condition (5) is surely *sufficient* for the map $A$ defining a projectively equivariant linear map. In the following, we will prove that in many important cases, it in fact also is *necessary*. To make our considerations general enough to include examples like Example 2.6, we will not assume that $\rho$ is a projection of a $\widehat{\rho}$, but only that we can 'lift' $\rho$ to a linear representation of a so-called covering group of $G$. We need the following two definitions.

**Definition 2.9.** (Hall, 2015, Def. 5.36) Let $G$ be a topological group. A group $H$ is called a *covering group* of $G$ if there exists a *group covering* $\varphi : H \to G$, i.e. a surjective continuous group homomorphism which maps some neighbourhood of the unit element $e_H \in H$ to a neighbourhood of the unit element $e_G \in G$ homeomorphically.

**Definition 2.10.** Let $\rho : G \to \mathrm{PGL}(V)$ be a projective representation, and $H$ a covering group of $G$, with group covering $\varphi$. A *lift* of $\rho$ is a linear representation $\widehat{\rho} : H \to \mathrm{GL}(V)$ with $\rho \circ \varphi = \Pi_{\mathrm{GL}(V)} \circ \widehat{\rho}$.

*Example* 2.11. If $\rho : G \to \mathrm{PGL}(V)$ is a projection of a linear representation $\widehat{\rho} : G \to \mathrm{GL}(V)$ as in Example 2.4, then $\widehat{\rho}$ is a lift of $\rho$ (the covering group is simply $G$ itself).

*Example* 2.12. The linear representations related to spin discussed in Example 2.6 are defined on $\mathrm{SU}(2)$, which is a covering group of $\mathrm{SO}(3)$. The linear representations are lifts of the projective representations of $\mathrm{SO}(3)$ discussed there.

*Remark* 2.13. Given a projective representation $\rho$ of a group $G$, there is a canonical way of constructing a covering group $H$ (dependent on $\rho$) and linear representation $\widehat{\rho}$ of $H$ which lifts $\rho$. This construction is however mainly of theoretical interest, and our subsequent results will not give us much insight when this lift is used. We discuss this further in Appendix B.7.

Let us first reformulate the equivariance problem slightly as is routinely done in the linear case (Weiler & Cesa, 2019; Finzi et al., 2021). Given projective representations $\rho_0$ and $\rho_1$ on $\mathrm{P}(V)$ and $\mathrm{P}(W)$, we may define a projective representation $\rho : G \to \mathrm{PGL}(\mathrm{Hom}(V, W))$ on the space $\mathrm{Hom}(V, W)$ of linear maps between $V$ and $W$, through

$$\rho(g)A = \rho_1(g)A\rho_0^{-1}(g).$$

On the left hand side we use $\rho(g)$ to transform $A \in \mathrm{P}(\mathrm{Hom}(V, W))$ to obtain a new element in $\mathrm{P}(\mathrm{Hom}(V, W))$. On the right hand side we write this new element out explicitly—it is a composition of the three maps $\rho_0^{-1}(g)$, $A$ and $\rho_1(g)$. Just as in the linear case, we can reformulate projective equivariance of an $A \in \mathrm{Hom}(V, W)$ as an *invariance* equation under $\rho$.

**Lemma 2.14.** *A linear map $A : V \to W$ is projectively equivariant if and only if $A$ is invariant under $\rho$, that is $\rho(g)A = A$ for all $g \in G$. These equations are understood in $\mathrm{P}(\mathrm{Hom}(V, W))$.*

The proof, which in contrast to the nonlinear case necessitates a non-trivial technical effort, can be found in the appendix. The key is that the equivariance condition for $A$ means that any $x$ solves the 'eigenvalue problem' $\rho_1(g)Ax = \lambda_{x,g}A\rho_0(g)x$, from which $\rho(g)A = A$ in $\mathrm{P}(\mathrm{Hom}(V,W))$ can be deduced. The lemma implies that we can concentrate on invariance equations

$$\rho(g)v = v \text{ for all } g \in G \tag{$\mathrm{Proj}_G$}$$

which, importantly, are understood in $\mathrm{P}(\mathcal{V})$ for some general vector space $\mathcal{V}$ (which covers the case $\mathcal{V} = \mathrm{Hom}(V,W)$). With this knowledge, the main result of this section is relatively easy to show. It says that given a lift $\widehat{\rho}$ of $\rho$, the solutions of projective invariance equations $(\mathrm{Proj}_G)$ of $\rho$ are exactly the ones of the $\varepsilon$-linear invariance equations of $\widehat{\rho}$ w.r.t. the covering group $H$,

$$\widehat{\rho}(h)v = \varepsilon(h)v \text{ for all } h \in H, \tag{$\mathrm{Lin}_H^\varepsilon$}$$

where $\varepsilon$ ranges the whole of $H^*$. Note in particular that $(\mathrm{Lin}_H^1)$ is nothing but the standard linear invariance problem for $\widehat{\rho}$. For a given $\varepsilon \in H^*$, we denote the space of solutions to $(\mathrm{Lin}_H^\varepsilon)$ by $U^\varepsilon$.

**Theorem 2.15.** *Let $G$ be a group and $H$ a covering group of $G$ with covering map $\varphi$. Further, let $\rho: G \to \mathrm{PGL}(\mathcal{V})$ be a projective representation of $G$ and $\widehat{\rho}: H \to \mathrm{GL}(\mathcal{V})$ a lift of $\rho$. Then, the following are equivalent*

$$(i) \; v \text{ solves } (\mathrm{Proj}_G). \qquad (ii) \; v \text{ is the equivalence class of some } x \in U^\varepsilon, \; \varepsilon \in H^*.$$

*Proof.* To prove that $(ii) \Rightarrow (i)$, note that $\Pi_{\mathcal{V}}(Mv) = \Pi_{\mathrm{GL}(\mathcal{V})}(M)\Pi_{\mathcal{V}}(v)$ for $M \in \mathrm{GL}(\mathcal{V})$ and $v \in \mathcal{V}$. Let $x$ solve $(\mathrm{Lin}_H^\varepsilon)$. For any $g \in G$, there is a $h$ such that $\varphi(h) = g$ and hence

$$\rho(g)\Pi_{\mathcal{V}}(x) = \rho(\varphi(h))\Pi_{\mathcal{V}}(x) = \Pi_{\mathrm{GL}(\mathcal{V})}(\widehat{\rho}(h))\Pi_{\mathcal{V}}(x) = \Pi_{\mathcal{V}}(\widehat{\rho}(h)x) = \Pi_{\mathcal{V}}(\varepsilon(x)x) = \Pi_{\mathcal{V}}(x),$$

meaning that the equivalence class of $x$ solves $(\mathrm{Proj}_G)$.

To prove that $(i) \Rightarrow (ii)$, let $\Pi_{\mathcal{V}}(x)$ be a solution of $(\mathrm{Proj}_G)$ for some $x \neq 0$. $\widehat{\rho}(h)x$ must then for all $h$ lie in the subspace spanned by $x$—in other words, there must exist a map $\lambda: H \to \mathbb{F}$ with $\widehat{\rho}(h)x = \lambda(h)x$. Since $\lambda(h)\lambda(k)x = \widehat{\rho}(h)\widehat{\rho}(k)x = \widehat{\rho}(hk)x = \lambda(hk)x$, $\lambda$ must be a group homomorphism. Also, since $\widehat{\rho}$ is continuous, $\lambda$ must also be. That is, $\lambda \in H^*$, and $x \in U^\lambda$. $\square$

## 2.2 The structure of the spaces $U^\varepsilon$

Let us study the spaces $U^\varepsilon$. Before looking at important special cases, which will reveal the important consequences of Theorem 2.15 discussed in the introduction, let us begin by stating two properties that hold in general.

**Proposition 2.16.** *(i) If $\mathcal{V}$ is finite dimensional and $H$ is compact, $U^\varepsilon$ is only non-trivial if $\varepsilon$ maps into the unit circle.*

*(ii) The $U^\varepsilon$ are contained in the space $U_{HH}$ of solutions of the linear invariance problem*

$$\widehat{\rho}(h)v = v \text{ for all } h \in \{H, H\}, \tag{$\mathrm{Lin}_{\{H,H\}}^1$}$$

*of the* commutator subgroup *$\{H, H\}$, i.e. the subgroup generated by commutators $\{h, k\} = hkh^{-1}k^{-1}, h, k \in H$. If $\mathbb{F} = \mathbb{C}$, $\mathcal{V}$ is finite dimensional and $H$ is compact, $U_{HH}$ is even the direct sum of the $U^\varepsilon, \varepsilon \in H^*$.*

The proofs require slightly more involved mathematical machinery than above. The idea for $(i)$ is that the $\widehat{\rho}$ in this case can be assumed unitary. The "$\subseteq$" part of $(ii)$ is a simple calculation, whereas we for the "direct sum" part need to utilize the simultaneous diagonalization theorem for commuting families of unitary operators. The details are given in Appendix B.4.

We move on to discuss the structure of the spaces $U^\varepsilon$ for three relevant cases. We will make use of a few well-known group theory results — for convenience, we give proofs of them in B.8.

**The rotation group.** $\mathrm{SO}(3)$ is a *perfect* group, meaning that $\mathrm{SO}(3)$ is equal to its commutator subgroup $\{\mathrm{SO}(3), \mathrm{SO}(3)\}$. This has the consequence that $\mathrm{SO}(3)^*$ only contains the trivial character $1$, which trivially shows that $(\mathrm{Proj}_{\mathrm{SO}(3)})$ is equal to the space $U^1$. In other words, we have the following corollary.

**Corollary 2.17.** *For projections of linear representations* $\widehat{\rho} : \mathrm{SO}(3) \to \mathrm{GL}(\mathcal{V})$*, (*$\mathrm{Proj}_{\mathrm{SO}(3)}$*) is equivalent to the linear equivariance problem* ($\mathrm{Lin}^1_{\mathrm{SO}(3)}$)

This means that if one wants to construct a canonical projectively equivariant neural net (Example 2.1), one either has to settle with the already available linearly equivariant ones, or construct a non-linearity that is projectively, but not linearly equivariant. To construct such non-linearities is interesting future work, but unrelated to the linear equivariance questions we tackle here, and hence out of the scope of this work.

The pinhole camera representation is a projection of a linear representation of $\mathrm{SO}(3)$. Not all projective representations are, though —they can also be linear projections of representations of $\mathrm{SU}(2)$. Since $\mathrm{SU}(2)$ also is perfect, there is again an equivalence, but only on the $\mathrm{SU}(2)$-level. Hence, in general *the projective invariance problem for the $\mathrm{SO}(3)$-representation is not necessarily equivalent to a linear equivariance problem* ($\mathrm{Lin}^1_{\mathrm{SO}(3)}$)*, but always to one of the form* ($\mathrm{Lin}^1_{\mathrm{SU}(2)}$).

**The permutation group.** The behaviour of $\mathrm{SO}(3)$ is in some sense very special. More specifically, it can be proven that if $G$ is compact, $G^* = \{1\}$ only when $G$ is perfect (see Section B.8). However, equivalence of ($\mathrm{Proj}_{\mathrm{SO}(3)}$) and ($\mathrm{Lin}^\varepsilon_H$) can still occur when $G^*$ is not trivial.

An example of this is the permutation group $S_n$ acting on spaces of tensors. The character group $S_n^*$ of the permutation group contains two elements: 1 and the sign function sgn. This means that the solutions to ($\mathrm{Proj}_{S_n}$) form two subspaces $U^1$ and $U^{\mathrm{sgn}}$. However, in the important case of $\rho$ being a projection of the canonical representation $\widehat{\rho}$ of $S_n$ on the tensor space $(\mathbb{F}^n)^{\otimes k}$ defined by $(\widehat{\rho}(\pi)x)_I = x_{\pi^{-1}(I)}$ for any multi-index $I \in [n]^k$ , something interesting happens: $U^{\mathrm{sgn}}$ is empty for $n \geq k + 2$. We prove this in Section B.5. In particular, we can draw another interesting corollary.

**Corollary 2.18.** *For the canonical action of $S_n$ on $\mathcal{V} = (\mathbb{F}^n)^k$ for $n \geq k + 2$, (*$\mathrm{Proj}_{S_n}$*) is equivalent to the linear equivariance problem* ($\mathrm{Lin}^1_{S_n}$).

In more practical terms, this means that unless tensors of very high order ($k \leq n - 2$) are used, we cannot construct canonical architectures operating on tensors that are projectively, but not linearly, equivariant to permutations without designing non-standard equivariant non-linearites.

**The translation group.** $\mathbb{Z}_n^*$ is isomorphic to the set of $n$:th roots of unity: every root of unity $\omega$ defines a character through $\varepsilon_\omega(k) = \omega^k, k \in \mathbb{Z}_n$. Furthermore, for the standard linear representation on $\mathbb{F}^n$, each of the spaces $U^{\varepsilon_\omega}$ is non trivial—it contains the element defined through $v_k = \omega^{-k}$: $(\rho(\ell)v)_k = v_{k-\ell} = \omega^{\ell-k} = \varepsilon_\omega(\ell)v_k$. In the case $\mathbb{F} = \mathbb{C}$, this leaves us with $n$ linear spaces of protectively invariant elements—which are not hard to identify as the components of the Fourier transform of $v$. If $\mathbb{F} = \mathbb{R}$, the number of roots of unity depend on $n$ — if $n$ is odd, there are two roots $\{+1, -1\}$, but if $n$ is even, the only root is $+1$. In the latter case, we thus again have an equivalence between ($\mathrm{Proj}_{\mathbb{Z}_n}$) and ($\mathrm{Lin}^1_{\mathbb{Z}_n}$).

## 3   A projectively equivariant architecture

Theorem 2.15 is mainly a result which restricts the construction of equivariant networks. Still, we can use its message to construct a new, projectively equivariant architecture as follows. We describe the architecture as a series of maps $V_k \to V_{k+1}$ between finite-dimensional spaces $V_k$. Theorem 2.15 states that all projectively equivariant linear maps $V_0 \to V_1$ lie in some space $U^\varepsilon$. Hence, for each $\varepsilon \in H^*$, we may first multiply the input features $v \in V_0$ with maps $A_0^\varepsilon \in U^\varepsilon$, to form new, projectively equivariant features. By subsequently applying an equivariant non-linearity $\sigma_0 : V_1 \to V_1$, we obtain the $V_1$-valued features

$$v_1^\varepsilon = \sigma_0(A_0^\varepsilon v), \quad A_0^\varepsilon \in W_0^\varepsilon, \varepsilon \in H^*. \tag{6}$$

It is not hard to see that as soon as the non-linearity $\sigma_0$ commutes with the elements $\varepsilon(h)$, $h \in H$, the features $v_1^\varepsilon$ will be projectively equivariant. Note that by Proposition 2.16, since the $V_i$ are finite dimensional and $H$ is compact, the values $\epsilon(g)$ will always lie on the unit circle. Hence, we can use tanh-nonlinearity in the real case, or the `modReLU` (Arjovsky et al., 2016) in the complex-valued one.

Also note that we here get one feature for each $\varepsilon \in H^*$, so we map from $V_0$ to a $|H^*|$-tuple of $V_1$'s. $U^\varepsilon$ is however nontrivial only for finitely many $\varepsilon$ – this is a consequence of Proposition 2.16 and the finite-dimensionality of

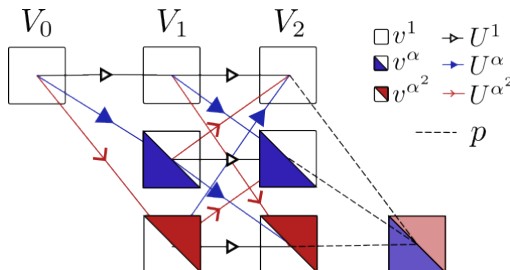

Figure 2: The structure of a three-layer projectively equivariant net for $H = \mathbb{Z}_3$. $\mathbb{Z}_3^*$ consists of three elements of the form $\epsilon(k) = \theta^k$, where $\theta = 1, \alpha, \alpha^2$, $\alpha = e^{2\pi i/3}$—we write $\mathbb{Z}_3^* = \{1, \alpha, \alpha^2\}$. The features $v^\delta$ are combined with linear operators in $U^\gamma$ to produce new features $v^\varepsilon$, with $\gamma, \delta, \epsilon \in \mathbb{Z}_3^*$. In a final step, the features of the $\varepsilon$-indexed tuples are linearly combined with a 'selector' $p$ to yield one single output. Best viewed in color.

the $V_k$. Said proposition also shows that the sum of their dimensions does not exceed $\dim(U_{HH})$. Hence, each tuple $(A_k^\varepsilon)_{\varepsilon \in H^*}$ can be represented with no more than $\dim(U_{HH}) \leq \dim(V_k) \cdot \dim(V_{k+1})$ scalars.

The features $v_1^\varepsilon$ can be combined with new linear maps $A_1^\gamma$, $\gamma \in H^*$ to form new features as follows:

$$v_2^\varepsilon = \sigma_1 \left( \sum_{\gamma, \delta \in H^*, \gamma\delta=\varepsilon} A_1^\gamma v_1^\delta \right). \tag{7}$$

This process can continue for $k = 3, 4, \ldots, K-1$ until we arrive at projectively equivariant $V_K$-valued features. We record that as a theorem, which we formally prove in Appendix B.

**Theorem 3.1.** *If the $\sigma_i$ commute with the elements $\varepsilon(g)$, $\varepsilon \in H^*$, $h \in H$, the features $v_k^\varepsilon$ are projectively equivariant.*

Also, the above construction inherently produces $|H^*|$ output features $v_K^\varepsilon \in V_K$. If we want a single output feature, we propose to learn a 'selection-vector' $p \in \mathbb{F}^{|H^*|}$, that linearly decides which feature(s) to 'attend' to:

$$w = \sum_{\varepsilon \in H^*} p_\varepsilon v_K^\varepsilon.$$

Technically, this does not yield an exactly equivariant function unless all but one entry in $p$ is equal to zero. Still, we believe it is a reasonable heuristic. In particular, we cannot a priori pick one of the $v_K^\varepsilon$ as output, since that fixes the linear transformation behaviour of the network. For instance, choosing $v_K^1$ leads to a linearly equivariant network, not able to capture non-trivial projective equivariance. Note also that we may employ sparse regularization techniques to encourage sparse $p$, e.g. to add an $\ell_1$-term to the loss. We provide a graphical depiction of our approach for the (simple but non-trivial) case of $H = \mathbb{Z}_3$ in Figure 2.

### 3.1 Relation to earlier work

As described above, once we fix $\varepsilon$, (5) is a linear equivariance condition on the linear map $A$. Hence the construction proposed here is equivalent to the following linear equivariance approach. For each feature space $V_k$, $k > 0$, duplicate it $|H^*|$ times to form a new feature space $\widetilde{V}_k = \bigoplus_{\varepsilon \in H^*} V_k$ and select the linear representation $\bigoplus_{\varepsilon \in H^*} \widehat{\rho}^\varepsilon$ to act on $\widetilde{V}_k$. Then parametrize the neural network by linearly equivariant maps $\widetilde{V}_k \to \widetilde{V}_{k+1}$. Depending on the group $H$ and the spaces $\widetilde{V}_k$ this can be done by various methods found in the literature (Cohen & Welling, 2016; Finzi et al., 2021; Cesa et al., 2021; Weiler & Cesa, 2019). In particular, if $U^\varepsilon$ is empty for $\varepsilon \neq 1$, or if there are no non-trivial $\varepsilon$, this construction will be exactly the same as the standard linearly equivariant one. This is for instance always the case for $H = SO(3)$ as explained leading up to Corollary 2.17.

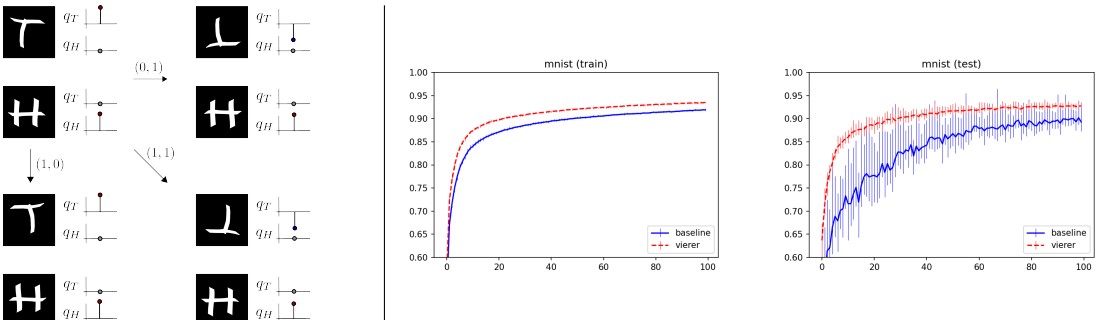

Figure 3: Left: The group action of the example in Section 3.2. Right: Training (left) and test (right) accuracy for the MNIST experiments. The lines report the median performance for each epoch, wheras the errorbars depict confidence intervals of 80%.

## 3.2 A possible application: Class-dependent symmetries

Let us sketch a somewhat surprising setting where our projectively equivariant network can be applied: Classification problems which exhibit *class-dependent* symmetries.

For clarity, let us consider a very concrete example: The problem of detecting 'T'-shapes in natural images. We may phrase this as learning a function $p_T : \mathbb{R}^{n,n} \to [0,1]$ giving the probability that the image $v \in \mathbb{R}^{n,n}$ contains a 'T'. This probability is not changed when the image is translated, or when it is horizontally flipped. The same is not true when flipping the image vertically—an image containing a 'T' will instead contain a '$\perp$' symbol. In other words, $p_T(v) \approx 1 \Rightarrow p_T(v_{\text{ver. flip}}) \approx 0$. The action of the two flips, which commute, can be modelled with the help of *Klein's Vierergroup*[1] $\mathbb{Z}_2^2$—$(1,0)$ corresponds to the vertical flip, and $(0,1)$ to the horizontal one.

Let us write $p_T = p_0 e^{q_T}$, where $p_0$ is some small number. If $v$ is an image neither containing a 'T' or a '$\perp$', $p_T(v)$ should be small — letting $q_T(v) = 0$ suffices in this case. If $v$ instead is an image containing a 'T', $q_T(v)$ should be large, and $q_T(v_{\text{ver. flip}})$ should be very small. This is satisfied by functions $q$ with $q(v_{\text{ver. flip}}) = -q(v)$. Hence, $q_T$ should modulo a multiplication with $-1$ not change by any flip — that is, $q_T$ should be projectively invariant. Note that 'projectively invariant' here a priori is too unrestrictive — it formally means $q_T(v_{\text{ver. flip}}) = \lambda q_T(v)$ for arbitrary $\lambda \neq 0$, which any function that is non-zero for both $v_{\text{ver. flip}}$ and $v$ satisfies. As we have seen, in our architecture, the scalar $\lambda$ can however only be equal to $\pm 1$, which is the behaviour we search for.

The exact same reasoning can be applied to the probability $p_H$ of an image containing a 'H'. In this case, however, $p_H$ — or equivalently, $q_H$ — is linearly, and therefore projectively, invariant of either flip. Put differently, $q_H$ and $q_T$ are linearly equivariant with respect to different representations $\rho^\varepsilon$, and therefore both projectively equivariant w.r.t. the same projective representation $\rho$. See also Figure 3.

Now imagine training both $q_H$ and $q_T$ on a dataset containing both 'T':s and 'H':s. If we were to use a canononical linearly equivariant architecture (with respect to a single $\rho^\varepsilon$), we could not expect a good performance. Indeed, one of the functions would necessarily not have the proper transformation behaviour. In contrast, our architecture has the chance — through learning class-dependent selection vectors — to adapt to the correct symmetry for the different classes.

*Remark* 3.2. In this particular example, since we know the symmetries of each class a-priori, we could of course choose two *different* linearly equivariant architectures to learn each function. Our architecture does however not need this assumption – it can be applied also when the symmetries of the individual classes are not known a-priori.

As a proof of concept, we perform a toy experiment in a similar setting to the above. We will define an image classification problem which is linearly invariant to translations, and projectively invariant to flips along the

---

[1]'Vier' is German for 'four', referring to the number of elements in the group.

horizontal and vertical axes, i.e., the Vierergroup $\mathbb{Z}_2^2$. Its character group also contains four elements—the values in $(1,0)$ and $(0,1)$ can both be either $+1$ or $-1$.

**Data.** We modify the MNIST dataset (LeCun et al., 1998), by adding an additional class, which we will refer to as 'NaN'. When an image $v \in \mathbb{R}^{n,n}$ is loaded, we randomly (with equal probability) either use the image as is, flip it horizontally or flip it vertically. When flipped, the labels are changed, but differently depending on the label: if it is either 0, 1 or 2, it stays the same. If it is 3, 4 or 5, we change the label to NaN if the image is flipped horizontally, but else not. If it is 6 or 7, we instead change the label if the image is flipped vertically, but else not. The labels of 8 and 9 are changed regardless which flip is applied. Note that these rules for class assignments are completely arbitrarily chosen, and we do not use the knowledge of them when building our model.

**Models.** Since we operate on images, we build a linearly translation-equivariant architecture by choosing layers in the space of $3 \times 3$-convolutional layers. We create **ViererNet**—a four-layer projectively equivariant network according to the procedure described in this section—and a four-layer baseline CNN with comparable number of parameters. Details about the models can be found in Appendix D.

**Results.** We train 30 models of each type. The evolution of the test and training accuracy is depicted in Figure 3, along with confidence intervals containing 80% of the runs. We clearly see that the ViererNet is both better at fitting the data, and is more stable and requires shorter training to generalize to the test-data. To give a quantitative comparison, we for each model determine which epoch gives the best median performance on the test data. The ViererNet then achieves a median accuracy of 92.8%, whereas the baseline only achieves 90.2%. We subsequently use those epochs to test the hypothesis that the ViererNet outperforms the baseline. Indeed, the performance difference is significant ($p < 0.025$).

In Appendix D, similar modifications of the CIFAR10 dataset are considered. This dataset is less suited for this experiment (since one of the flips often leads to a plausible CIFAR10-image), and the results are less clear: The baseline outperforms the ViererNet slightly, but not significantly so. To some extent, the trend that the ViererNet requires less training to generalize persists.

## 4 Generalizing Tensor Field Networks to projective representations of $\mathrm{SO}(3)$

In this section we consider a regression task on point clouds. The task is equivariant under a projective representation of $\mathrm{SO}(3)$ corresponding to a linear representation of $\mathrm{SU}(2)$ as was briefly described in Example 2.6. We give an introduction to the representation theory of $\mathrm{SU}(2)$ in Appendix C. In short, $\mathrm{SU}(2)$ has a $2\ell + 1$ dimensional irreducible representation for each $\ell = 0, 1/2, 1, 3/2, \ldots$. The vector spaces these representations act on are isomorphic to $\mathbb{C}^{2\ell+1}$ and we denote them by $\mathcal{V}_\ell$.

Our main result (Theorem 2.15) and the discussion in Section 2.2 show that we for projective $\mathrm{SO}(3)$ equivariance could build a net using linearly $\mathrm{SU}(2)$ equivariant layers. We will here use linear layers of the tensor product type used in Tensor Field Networks (TFNs) (Thomas et al., 2018) and many subsequent works (Geiger & Smidt, 2022; Brandstetter et al., 2021). The filters and features are then functions on $\mathbb{R}^3$, and the resulting construction hence deviates slightly from our theoretical results – see Appendix C – but we think that generalizing this canonical approach for point cloud processing networks to projective equivariance is of high relevance to the paper.

**Data.** The data in the experiment consists of point clouds equipped with spinor features and regression targets that are spinors. By spinors we mean elements of $\mathcal{V}_{1/2}$. That is, one data sample is given by $(\{(x_i, s_i)\}_{i=1}^n, t)$ where $x_i \in \mathbb{R}^3$ are the 3D positions of the points, $s_i \in \mathcal{V}_{1/2} \sim \mathbb{C}^2$ are the attached spinor features and $t \in \mathcal{V}_{1/2}$ is the regression target. We define an action of $\mathrm{SO}(3)$ on these samples by

$$R \cdot (\{(x_i, s_i)\}_{i=1}^n, t) := (\{(Rx_i, U(R)s_i)\}_{i=1}^n, U(R)t). \tag{8}$$

Here $U(R)$ is an element of $\mathrm{SU}(2)$ that corresponds to $R$. We note that since $\mathrm{SU}(2)$ double covers $\mathrm{SO}(3)$, $U(R)$ is only defined up to sign. Hence, $s_i$ and $t$ can be thought of as $\mathbb{R}$-projective, and the action of $\mathrm{SO}(3)$ on them as a projective representation.

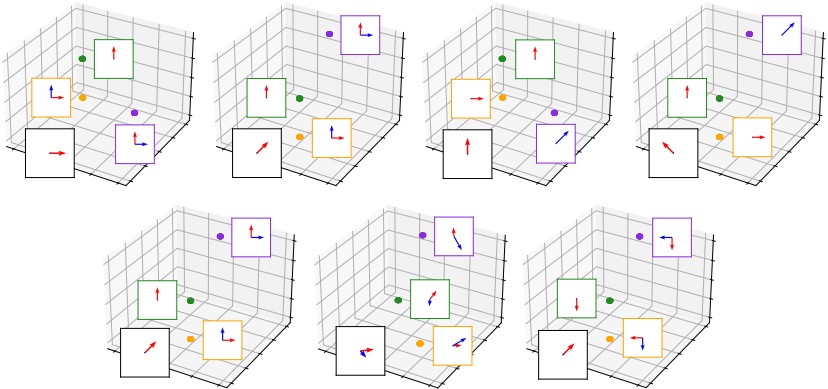

Figure 4: Data used for the regression task with SO(3)-equivariance. We define four classes, each class being a point cloud with three points in 3D. To each 3D point we attach a spinor feature $s \in \mathbb{C}^2$. These spinor features are plotted inside the frames next to each 3D point, the frames and points are color coded. $\mathrm{Re}(s)$ is plotted in red and $\mathrm{Im}(s)$ in blue. For every class a different spinor should be regressed, which is shown in the black frame. **Top:** The four classes. They are defined such that if one looks only at the spatial locations of the points, the first and third classes are equivalent as are the second and fourth. If one instead looks only at the spinor features, the first and second classes are equivalent as are the third and fourth. A network that regresses the output spinor correctly for all classes must take into account both the locations and the spinor features of each point. **Bottom:** The regression should be equivariant under rotations of the point clouds. When a point cloud is rotated by $R \in \mathrm{SO}(3)$, the spinor features and labels are transformed by a corresponding matrix $U(R) \in \mathrm{SU}(2)$. A random rotation of the leftmost point cloud is shown in the center column. Given a rotation $R$, the corresponding $U(R)$ is however only defined up to sign. E.g., the identity rotation corresponds to both the identity matrix $I \in \mathrm{SU}(2)$ and $-I$. The rightmost column illustrates a correct regression even though all spinor input features have been multiplied by $-I$, while the output spinor has not.

The task is to regress the target in a way that is equivariant to (8) and invariant to translations of the point clouds. The data is defined by four prototype samples that are shown in Figure 4. To make the task more difficult we also add Gaussian noise to the spatial coordinates $x_i$, varying the noise level (i.e., std.) from 0 to 2/5. During training, a network sees only the four prototypes + noise, while evaluation is done on rotated prototypes + noise. According to Theorem 2.15, and the discussion regarding the rotation group in Section 2.2, the projectively SO(3)-equivariant network architectures are in this case exactly the standard linearly SU(2)-equivariant networks. We illustrate the data and how it transforms under SO(3) in Figure 4.

**Models.** We will tackle the problem by building an equivariant net in the spirit of Tensor Field Networks (TFNs) (Thomas et al., 2018). Let us first recall how TFNs work. Our data are point clouds $\{x_i\}_{i=1}^n$ in $\mathbb{R}^3$ with features $f_i$ in some vector space $V$ attached to each point, on which SO(3) is acting through $\widehat{\rho}_V$. We map the inputs to point clouds with features $f_i'$ in some vector space $W$ (on which SO(3) acts through $\widehat{\rho}_W$) via convolution with a filter function $\Psi : \mathbb{R}^3 \to \mathrm{Hom}(W, V)$ through

$$f_i' = \sum_{j \neq i} \Psi(x_j - x_i) f_j. \tag{9}$$

Provided the filter $\Psi$ satisfies the invariance condition $\Psi(Rx) = \widehat{\rho}_W(R)\Psi(x)\widehat{\rho}_V(R)^{-1}$, $R \in \mathrm{SO}(3)$, this defines a layer invariant to translations and rotations of $\mathbb{R}^3$, and the SO(3)-action on $V$ and $W$. We could make an ansatz of $\Psi$ as a member of some finite-dimensional space of functions $\mathbb{R}^3 \to \mathrm{Hom}(V, W)$, and then resolve this invariance condition directly using our theory. However, the resulting arcitecture would be numerically heavy to implement. We have therefore instead used the idea of TFNs.

The idea of a TFN is to define $\Psi$ using a tensor product, $\Psi(x)v = \psi(x) \otimes v$, $v \in V$. Here, $\psi(x)$ lives in some third vector space $U$, and satisfies the condition $\psi(Rx) = \rho_U(R)\psi(x)$, $R \in \mathrm{SO}(3)$. $\Psi$ maps equivariantly into the space $W = U \otimes V$, which can be decomposed into spaces isomorphic to the irreps $\mathcal{V}_0, \mathcal{V}_1, \mathcal{V}_2, \ldots$ of the

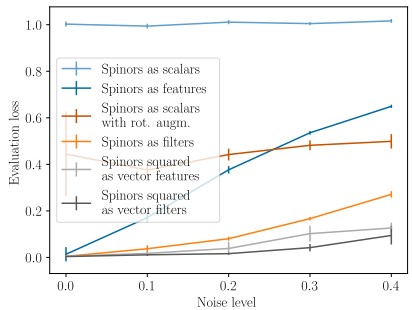

Figure 5: Results of the spinor regression experiments for varying architectures and spatial noise levels. We plot the mean over 30 runs, showing the standard deviation with errorbars.

action of SO(3). In this manner, we may transform features in different irreps into new ones living in other irreps. For instance, if we combine features and filters living in the space $\mathcal{V}_1 \sim \mathbb{R}^3$ (i.e., vectors), we end up with one scalar component $\mathcal{V}_0$, one vector component in $\mathcal{V}_1$ and a traceless symmetric $3 \times 3$-matrix component in $\mathcal{V}_2$. The higher order component $\mathcal{V}_2$ can be discarded to keep the feature dimensions low. Another type of layer are self-interaction layers where features of the same irrep at each point are linearly combined using learnt weight matrices.

The idea of our approach is now to let $U$ and $V$ be spaces on which SU(2) is acting, i.e., also consider features and vectors of half-integer spin. We limit ourselves to $\ell = 0, 1/2$ and 1, i.e., scalars, spinors and vectors, and call the resulting nets Spinor Field Networks. The hyperparameter choices for a Spinor Field Network in our implementation are the number of scalar, spinor and vector features output by each layer. We use sigmoid-gated nonlinearities (Weiler et al., 2018) for non-scalar features and `GeLU` for scalar features.

We consider in total five different models. (i) *Spinors as scalars.* A non-equivariant baseline TFN that treats input and output spinors as four real scalars. It has the correct equivariance for the 3D structure of the locations but not for the spinors. (ii) *Spinors as features.* An SU(2)-equivariant net as described above, with intermediate scalar,vector and spinor features. (iii) *Spinors squared as vector features.* Prior to the first layer, we tensor square the spinor features $s_i \mapsto s_i \otimes s_i$ and extract a vector feature at each point, which we then process SO(3)-equivariantly. This allows the network to only use real valued (scalar and vector) features in the intermediate layers, which is likely an advantage in most modern machine learning frameworks (including PyTorch which we use). In the last layer, the scalar and vector features are tensored with the input spinors to produce an output spinor.

The last network types have filters that consist of not only $\mathcal{V}_\ell$ valued functions, but also the input spinors themselves. That is, we define layers $f'_i = \sum_{j \neq i} (\psi(x_j - x_i) \oplus s_j) \otimes f_j$. This could be generalized to tensor spherical harmonics valued filters which we explain in C.1. We consider two such types: (vi) *Spinors as filters* works as just described. (v) *Spinors squared as vector filters* analogously defines vector valued filters by tensor squaring the input spinors as in the *Spinors squared as vector features* case. This again allows for a mostly real valued network.

Each network consists of three layers and the hyperparameters are chosen to make the number of parameters approximately equal between all networks. In all networks the input feature at a location $x_i$ consists of the average of the vectors $x_i - x_j$ for $j \neq i$ and for all except the two last networks also the spinor feature $s_i$ (interpreted as either 4 scalars, a spinor or a vector after squaring). The final output is the mean of the features at each input location, meaning that the last layer maps to a single spinor feature—except in the non-equivariant variant in which the last layer maps to 4 scalar features. We describe the training setup and further details in Appendix C.2.

**Results.** All networks are trained on just the four point clouds shown in the left part of Figure 4 (plus added spatial noise), except one version of the non-equivariant net which is trained on randomly rotated versions of these four point clouds as data augmentation. The evaluation is always on randomly rotated versions of the data. We present results in Figure 5. Notably, the networks that tensor square the spinors before the first layer to then work on real-valued data until the last layer perform the best. At low noise levels, the networks that work directly on complex-valued spinors also perform well, but their performance

degrades quickly with added noise. The network without correct equivariance unsurprisingly performs quite poorly with, and extremely poorly without, data augmentation. The networks with filters defined using the input spinors outperform the networks where the inputs are only fed in once at the start of the network. This is likely due to the fact that inserting the inputs in each layer makes the learning easier for the network.

## 5 Conclusion

In this paper, we theoretically studied the relation between projective and linear equivariance for neural network linear layers revealed that our proposed approach is the most general one. We in particular found examples (including SO(3), or rather SU(2)) for which the two problems are, somewhat surprisingly, equivalent. Building on the knowledge of the structure of the set of projectively equivariant layers our main result gave us, we proposed a neural network model for capturing projective equivariance, and tested it on a toy task with class-dependent symmetries. Finally, we experimentally evaluated the merit of using projective equivariance for tasks involving spinor-valued features.

**Acknowledgement**

All authors were supported by the Wallenberg AI, Autonomous Systems and Software Program (WASP) funded by the Knut and Alice Wallenberg Foundation. The computations were enabled by resources provided by the National Academic Infrastructure for Supercomputing in Sweden (NAISS) and the Swedish National Infrastructure for Computing (SNIC) at C3Se Chalmers, partially funded by the Swedish Research Council through grant agreements no. 2022-06725 and no. 2018-05973.

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

# A  Technical definitions

Table 1 contains a symbol glossary for various symbols used in this work. The remainder of this section contains definitions used in the text, in particular a few that did not have room in the main text, compiled for the convenience of the reader.

**Definition A.1** (Group homomorphism)**.** Given two groups $G, H$, a group homomorphism from $G$ to $H$ is a map $a : G \to H$ that respects the group structure, i.e., such that $a(g_1 g_2) = a(g_1)a(g_2)$ for all $g_1, g_2$ in $G$.

**Definition A.2.** Given a vector space $V$ over a field $\mathbb{F}$, $\mathrm{P}(V)$ is the corresponding projective space consisting of equivalence classes under the relation

$$v \sim w \iff v = \lambda w \text{ for some } \lambda \neq 0 \tag{10}$$

**Definition A.3** (General linear group, Projective linear group)**.** The general linear group $\mathrm{GL}(V)$ of a vector space $V$ is the group of all invertible linear maps from $V$ to itself. The set of equivalence classes of $\mathrm{GL}(V)$ under the equivalence relation $A \sim B \iff A = \lambda \cdot B$ for a $\lambda \in \mathbb{F}\backslash\{0\}$ is the projective linear group $\mathrm{PGL}(V)$.

**Definition A.4** (Linear representation)**.** A *linear representation* of a group $G$ on a vector space $V$ is a group homomorphism $\widehat{\rho} : G \to \mathrm{GL}(V)$.

**Definition A.5** (Projective representation)**.** A *projective representation* of a group $G$ on a projective space space $\mathrm{P}(V)$ is a group homomorphism $\rho : G \to \mathrm{PGL}(V)$.

**Definition A.6** (Linear (projective) equivariance)**.** Let $V$ and $W$ be vector spaces equipped with linear (projective) representations $\varrho_V$ and $\varrho_W$. A linear map $A \in \mathrm{Hom}(V, W)$ is then *linearly (projectively) equivariant* if

$$A\varrho_V(g)v = \varrho_W(g)Av, \quad g \in G, v \in V$$

as elements in $W$ (in $\mathrm{P}(W)$).

**Definition A.7** (Topological group)**.** A topological group is a set $G$ that is simultaneously a topological space and a group such that

1. The group operation is continuous w.r.t. the topology.

2. The inversion map $g \mapsto g^{-1}$ is continuous w.r.t. the topology.

**Definition A.8** (Homeomorphism)**.** A homeomorphism is an invertible, continuous map between two topological spaces whose inverse is also continuous. A function maps a set homeomorphically if the restriction of the function to that set is a homeomorphism.

**Definition A.9** (Character group)**.** Let $G$ be a topological group. The set of continuous group homomorphisms $\varepsilon : G \to \mathbb{F}$ is called the *character group* of $G$, and is denoted $G^*$.

**Definition A.10** (Covering group)**.** Let $G$ be a topological group. A group $H$ is called a covering group of $G$ if there exists a *group covering* $\varphi : H \to G$, i.e. a surjective continuous group homomorphism which maps some neighbourhood of the unit element $e_H \in H$ to a neighborhood of the unit element $e_G \in G$ homeomorphically.

**Definition A.11** (Lift of a projective representation)**.** Let $\rho : G \to \mathrm{PGL}(V)$ be a projective representation, and $H$ a covering group of $G$, with group covering $\varphi$. A *lift* of $\rho$ is a linear representation $\widehat{\rho} : H \to \mathrm{GL}(V)$ with $\rho \circ \varphi = \Pi_{\mathrm{GL}(V)} \circ \widehat{\rho}$.

**Definition A.12** (Tensor product of finite dimensional vector spaces)**.** Given two vector spaces $U$ with basis $\{u_i\}_{i=1}^n$ and $V$ with basis $\{v_i\}_{i=1}^m$, their tensor product $U \otimes V$ is a vector space with one basis vector for each pair $(u_i, v_j)$ of one basis vector from $U$ and one from $V$. The basis vectors of $U \otimes V$ are typically denoted $\{u_i \otimes v_j\}_{i=1, j=1}^{n, m}$.

**Definition A.13** (Tensor product of vectors)**.** The tensor product of an element $u = \sum_i \alpha_i u_i$ in $U$ and an element $v = \sum_j \beta_j v_j$ in $V$ is the element $u \otimes v = \sum_{i,j} \alpha_i \beta_j u_i \otimes v_j$ of $U \otimes V$. This defines a bilinear map $(u, v) \mapsto u \otimes v$.

**Definition A.14** (Tensor power). The tensor power $U^{\otimes k}$ is simply the tensor product of $U$ with itself $k$ times.

**Definition A.15** (Irreducability). A linear representation $\widehat{\rho}: G \to \mathrm{GL}(V)$ is called *irreducible* if there is no subspace $U \notin \{\{0\}, V\}$ which is invariant under all $\widehat{\rho}(g)$, $g \in G$.

## B    Omitted proofs

In this section, we collect some further proofs we left out in the main text.

### B.1    Induced projected representations

We begin, for completeness, by remarking and proving a small statement that we use implicitly throughout the entire article.

**Proposition B.1.** *Let $G$ be a group, $V$ and $W$ vector spaces, $\widehat{\rho}_V$ and $\widehat{\rho}_W$ linear representations of $G$ on $V$ and $W$, respectively, and $\rho_V$ and $\rho_W$ their corresponding projected representations. Then, the induced projective representation $\rho$ on $\mathrm{Hom}(V, W)$*

$$\rho(g)A = \rho_W(g) \circ A \circ \rho_V(g)^{-1}$$

*is the projection of the corresponding induced linear projection*

$$\widehat{\rho}(g)A = \widehat{\rho}_W(g) \circ A \circ \widehat{\rho}_V(g)^{-1}.$$

*Proof.* What we need to prove is that if $M$ is equivalent to $\widehat{\rho}_W(g)$ and $N$ is equivalent to $\widehat{\rho}_V(g)$, the linear map

$$R_{MN} : \mathrm{Hom}(V, W) \to \mathrm{Hom}(V, W), A \mapsto MAN^{-1}$$

is equivalent to $\widehat{\rho}(g)$. However, the stated equivalences mean that $M = \lambda \widehat{\rho}_W(g)$ and $N = \mu \widehat{\rho}_V(g)$ for some non-zero $\mu, \lambda$. This in turn shows that

$$R_{MN}(A) = \lambda \widehat{\rho}_W(g) A (\mu \widehat{\rho}_V(g))^{-1} = \lambda \mu^{-1} \widehat{\rho}(g) A,$$

i.e., that $R_{MN}$ is equivalent to $\widehat{\rho}(g)$. The claim has been proven. $\square$

### B.2    Theorem 3.1

*Proof of Theorem 3.1.* We will prove that

$$v_k^\varepsilon(\widehat{\rho}_0(g)) = \varepsilon(g) \widehat{\rho}_k(g) v_k^\varepsilon(v)$$

for $g \in G$, $v \in V$ for all $k$. This in particular means $v_k^\varepsilon(\widehat{\rho}_0(g)) = \widehat{\rho}_k(g) v_k^\varepsilon(v)$ as elements in $\mathrm{P}(V_k)$, which is to be proven.

We proceed with induction starting with $k = 1$. We have for $g \in G$ and $\varepsilon \in G^*$ arbitrary

$$v_1^\varepsilon(\widehat{\rho}_0(g)v) = \sigma_0(\widehat{\rho}_1(g)\widehat{\rho}_1(g)^{-1}A_0^\varepsilon\widehat{\rho}_0(g)v) \overset{(5)}{=} \widehat{\rho}_1(h)\sigma_0(\varepsilon(g)A_0^\varepsilon v) = \varepsilon(g)\widehat{\rho}_1(g)v_1^\varepsilon(v),$$

where we in the final step used our assumption on $\sigma_0$.

To prove the induction step $k \to k+1$, let us first concentrate on each expression $A_k^\gamma v_k^\delta$ in (7). Due to the induction assumption, we have $v_k^\delta(\widehat{\rho}_0(g)v) = \delta(g)\widehat{\rho}_k(g)v_1^\delta(g)$ for $g \in G$, $\delta \in G^*$. Consequently

$$A_k^\gamma v_k^\delta(\widehat{\rho}_0(g)v) = \delta(g)A_k^\gamma\widehat{\rho}_k(g)v_1^\delta(v) = \delta^1(t)\widehat{\rho}_{k+1}(g)\widehat{\rho}_{k+1}(g)^{-1}A_k^\gamma\widehat{\rho}_k(h)v_1^\delta(v)$$

$$\overset{(5)}{=} \delta(g)\gamma(g)\widehat{\rho}_{k+1}(g)A_k^\gamma v_1^\delta(v)$$

Summing over all $\gamma, \delta \in G^*$ with $\gamma \cdot \delta = \varepsilon$ yields a feature that transforms the claimed way. This is not changed through an application of the equivariant $\sigma_k$ – since it by assumption commutes with $\varepsilon(g)$. The claim has been proven. $\square$

### B.3 Lemma 2.14

*Proof of Lemma 2.14.* Equivariance of the map $A$ means that for all $g \in G$ and $v \in \mathrm{P}(V)$, we have $\rho_1(g)Av = A\rho_0(g)v$. Since all $\rho_0(g)$ are invertible, this is equivalent to

$$\rho_1(g)A\rho_0(g)^{-1}v = Av \text{ for all } v \in V.$$

Now, the above equality is not an equality of elements in $W$, but rather of elements in $\mathrm{P}(W)$. That is, it says that all vectors $v \in V$ are solutions of the generalized eigenvalue problem $\rho_1(g)A\rho_0(g)^{-1}v = \lambda Av$. The aim is to show that this implies that $\rho_1(g)A\rho_0(g)^{-1} = \lambda A$ for some $\lambda \in \mathbb{F}$. To show this, we proceed in two steps.

**Claim** Let $M \in \mathrm{GL}(V)$. If every vector $v \in V$ is an eigenvector of $M$, $M$ is a multiple of the identity.

**Proof** Suppose not. Then, since all vectors are eigenvectors, there exists $v \neq w \neq 0$ and $\lambda \neq \mu$ with $Mv = \lambda v$, $Mw = \mu v$. Now, again since all vectors are eigenvectors of $M$, there must exist a third scalar $\sigma$ with $M(v + w) = \sigma(v + w)$. Now,

$$\sigma(v + w) = M(v + w) = Mv + Mw = \lambda v + \mu w \iff (\sigma - \mu)w = (\mu - \sigma)v.$$

Now, the final equation can only be true if $\lambda - \mu = \mu - \sigma = 0$, i.e., $\lambda = \sigma = \mu$, which is a contradiction.

**Claim** Let $E, F \in \mathrm{Hom}(V, W)$. If every $v \in V$ is a solution of the generalized eigenvalue problem $Ev = \lambda Fv$, $E$ is a multiple of $F$.

**Proof** First, $E$ restricted to $\ker F$ must be the zero map, since for all $v \in \ker F$, $Ev = \lambda Fv = \lambda \cdot 0 = 0$. Secondly, $F^\circ : \ker F^\perp \to \mathrm{ran}\, F$ is an isomorphism, so that the generalized eigenvalue problem of $Ev = \lambda Fv$ on $\ker F^\perp$ is equivalent to the eigenvalue problem of $E(F^\circ)^{-1} : \mathrm{ran}\, F \to \mathrm{ran}\, F$. By the previous claim, $E(F^\circ)^{-1} = \lambda\,\mathrm{id}$ for some $\lambda \in F$, i.e., $E = \lambda F^\circ$ on $\ker F^\perp$. Since both $E$ and $F$ additionally are equal to the zero map on $\ker F$, the relation $E = \lambda F$ remains true also there. The claim has been proven. $\square$

### B.4 Proposition 2.16

*Proof of Proposition 2.16.* Ad $(i)$: If $\mathcal{V}$ is finite-dimensional and $H$ is compact, it is well known that we (by modifying the inner product on $\mathcal{V}$) may assume that $\widehat{\rho}$ is unitary—for convenience, a proof is given in Section B.8. This means that $|v| = |\widehat{\rho}(g)v|$ for all $h \in H$. If now $v \in U^\varepsilon$ is non trivial, we have $0 \neq |v| = |\widehat{\rho}(h)v| = |\varepsilon(h)v|$, i.e., $|\varepsilon(h)| = 1$ for all $h \in H$.

Ad $(ii)$: Let $\varepsilon \in H^*$, $x \in U^\varepsilon$ $k, h \in H$ be arbitrary. Then

$$\widehat{\rho}(\{k, h\})x = \widehat{\rho}(k)\widehat{\rho}(h)\widehat{\rho}(k)^{-1}\widehat{\rho}(h)^{-1}x = \varepsilon(k)\varepsilon(h)\varepsilon(k)^{-1}\varepsilon(h)^{-1}x = x.$$

Since $\widehat{\rho}$ is a group isomorphism, this relation extends to $\widehat{\rho}(\ell)x = x$ for all $\ell \in \{H, H\}$. This proves the first part.

To prove the second claim, we will show that the operators $\widehat{\rho}(h)$, $h \in H$ restricted to $U_{HH}$ are commuting operators $U_{HH} \to U_{HH}$. Since we are in the case $\mathbb{F} = \mathbb{C}$, and we WLOG can assume that they are unitary, this means that they are simultaneously diagonalizable, i.e.,

$$\widehat{\rho}(h)x_i = \lambda_i(h)x_i,$$

for some basis $x_i$ of $U_{HH}$ and 'eigenvalue functions' $\lambda_i : H \to \mathbb{C}$. With the exact same argument as in the proof of the main theorem, we show that $\lambda_i \in H^*$. Hence, the $x_i$ are elements of subspaces $U^\varepsilon$, which shows that $U_{HH} \subseteq \mathrm{span}(U^\varepsilon, \varepsilon \in H^*)$. Since the other inclusion holds in general by what we just proved, the claim follows.

So let us prove that the $\widehat{\rho}(h)$ are commuting operators $U_{HH} \to U_{HH}$. Both of this follows from the following equality:

$$\widehat{\rho}(k)\widehat{\rho}(h) = \widehat{\rho}(kh) = \widehat{\rho}(khk^{-1}h^{-1}hk) = \widehat{\rho}(h)\widehat{\rho}(k)\widehat{\rho}(\{k^{-1}, h^{-1}\}).$$

To show that $\widehat{\rho}(h)$ are operators $U_{HH} \to U_{HH}$, we may now argue that $x \in U_{HH}$, and $k \in \{H, H\}$ and $h \in H$ are arbitrary, it follows

$$\widehat{\rho}(k)\widehat{\rho}(h)x = \widehat{\rho}(h)\widehat{\rho}(k)\widehat{\rho}(\{k^{-1}, h^{-1}\})x \overset{x \in U_{HH}}{=} \widehat{\rho}(h)x$$

which means that $\widehat{\rho}(h)x \in U_{HH}$. To prove that the maps commute on $U_{HH}$ is even easier: for $x \in U_{HH}$ and $k, h \in H$ arbitrary, we immediately deduce

$$\widehat{\rho}(k)\widehat{\rho}(h)x = \widehat{\rho}(h)\widehat{\rho}(k)\widehat{\rho}(\{k^{-1}, h^{-1}\})x = \widehat{\rho}(h)\widehat{\rho}(k)x.$$

The proof is finished. $\square$

### B.5 $U^{\mathrm{sgn}}$ for the standard action on $(\mathbb{F}^n)^{\otimes k}$

**Proposition B.2.** *Let $\widehat{\rho}$ be the standard representation of $S_n$ on $(\mathbb{F}^n)^{\otimes k}$. For $n \geq k + 2$, $U^\varepsilon$ is nontrivial only for $\varepsilon = 1$. The bound is tight.*

*Proof.* Let $T \in (\mathbb{F}^n)^{\otimes k}$ be an element of $U^{\mathrm{sgn}}$, and let $I = (i_0, \ldots, i_{k-1})$ be an arbitrary $k$-index. Since $n \geq k + 2$, there are at least two elements $(i, j) \notin \{(i_0, \ldots, i_{k-1})\}$. Consider the transposition $\tau$ of $i$ and $j$. Then, $\tau(i_\ell) = i_\ell$, $\ell \in [k]$, and $\mathrm{sgn}(\tau) = -1$. Consequently,

$$T_I = T_{\tau(I)} = (\widehat{\rho}(\tau)T)_I = \mathrm{sgn}(\tau)T_I = -T_I,$$

since $T$ was in $U^{\mathrm{sgn}}$. Consequently, $T_I = 0$. Since $I$ was arbitrary, $T$ must be trivial, and first part of the claim follows.

To show that the bound is tight, it is enough to construct a non-zero element in $(\mathbb{F}^{k+1})^{\otimes k} \cap U^{\mathrm{sgn}}$. Let $I_0 = (0, 1, 2, \ldots, k) \in [k+1]^k$. For each multi-index $I$, there are either multiple indices in $I$, or there exists a unique permutation $\pi_I \in S_{n+1}$ so that $I = \pi_I^{-1}(I_0)$. Due to the uniqueness, it is not hard to show that $\pi_{\sigma^{-1}(I)} = \pi_I \circ \sigma$, $\sigma \in S_n$. Let us now define a tensor $S$ through

$$S_I = \begin{cases} 0 & \text{if } I \text{ contains multiple indices} \\ \mathrm{sgn}(\pi_I) & \text{else.} \end{cases}$$

We claim that $S$ is the sought element. To show this, let $I \in [k+1]^k$ and $\sigma \in S_n$ be arbitrary. We have

$$(\widehat{\rho}(\sigma)S)_I = S_{\sigma^{-1}(I)}$$

Now, if $I$ contains multiple indices, $\sigma^{-1}(I)$ must also, so $S_{\sigma^{-1}(I)} = 0$, and $(\widehat{\rho}(\sigma)S)_I = 0$ for such indices. If $I$ does not contain multiple indices, we deduce

$$S_{\sigma^{-1}(I)} = \mathrm{sgn}(\pi_{\sigma^{-1}(I)}) = \mathrm{sgn}(\pi_I \circ \sigma) = \mathrm{sgn}(\pi_I)\mathrm{sgn}(\sigma) = \mathrm{sgn}(\sigma)S_I.$$

Since $I$ was arbitrary, we have shown that $\widehat{\rho}(\sigma)S = \mathrm{sgn}(\sigma)S$, which was the claim. $\square$

*Remark* B.3. In an earlier version of the manuscript, we proved the above proposition in a different, arguably more clumsy, manner. Since it provides a nice connection to the results in (Maron et al., 2019), we include a sketch of that argument also here.

First, we realise that $U^{\mathrm{sgn}}$ being trivial is the same as to say that $(\mathrm{Proj}_{S_n})$ is equivalent to the linear invariance problem $(\mathrm{Lin}^1_{S_n})$. The solutions of the latter are surely included in the former (by e.g. Theorem 2.15) However, Proposition 2.16(ii) also tells us that the solutions of $(\mathrm{Proj}_{S_n})$ are included in the solution space of $(\mathrm{Lin}^1_{\{S_n, S_n\}})$. The commutator $\{S_n, S_n\}$ is the alternating group $A_n$, i.e., the group of permutations of sign 1 (see Section B.8). Now, the linear invariance problem $(\mathrm{Lin}^1_{A_n})$ was treated in (Maron et al., 2019)—it was shown that for $n \geq k + 2$, the set of solutions of $(\mathrm{Lin}^1_{A_n})$ is exactly equal to the one of $(\mathrm{Lin}^1_{S_n})$ (in fact, the proof of that builds on the same idea as our proof of Proposition B.2). Hence, we conclude

$$\text{Sol. of } (\mathrm{Lin}^1_{S_n}) \subseteq \text{Sol. of } (\mathrm{Proj}_{S_n}) \subseteq \text{Sol. of } (\mathrm{Lin}^1_{\{S_n, S_n\}}) = \text{Sol. of } (\mathrm{Lin}^1_{A_n}) = \text{Sol. of } (\mathrm{Lin}^1_{S_n}).$$

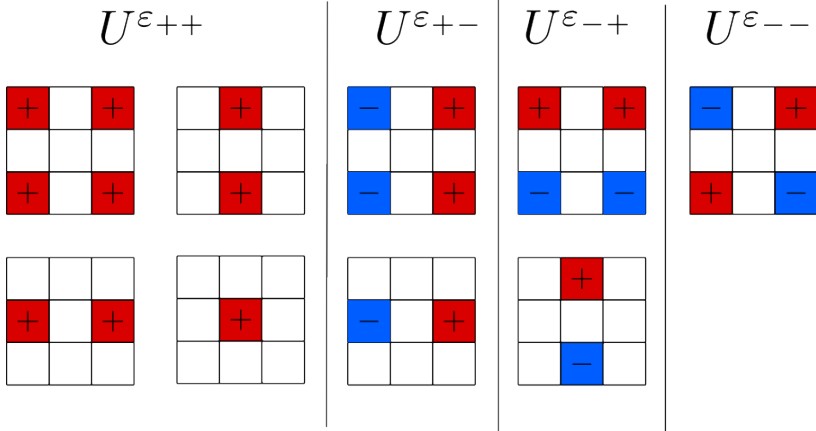

Figure 6: The bases for the spaces $U^\varepsilon$ for the action of $\mathbb{Z}_2^2$ on the space $C$ of $3 \times 3$ filters.

## B.6 The Vierergroup

Let us begin by describing the character group of $\mathbb{Z}_2^2$. $\mathbb{Z}_2$ is an abelian group generated by two elements, $(1, 0)$ and $(0, 1)$. Any group homomorphism $\varepsilon$ is hence uniquely determined by its values $\varepsilon((1, 0))$ and $\varepsilon((0, 1))$. Since $(1, 0) + (1, 0) = 0 = (0, 1) + (0, 1)$, the values must be square roots of one, i.e., either $+1$ or $-1$. Hence, $\mathbb{Z}_n^*$ contains four elements $\epsilon_{++}$, $\epsilon_{+-}$, $\epsilon_{-+}$, $\varepsilon_{--}$, defined through the following table:

| $(\mathbb{Z}_2^2)^* \setminus \mathbb{Z}_2^2$ | $(0, 0)$ | $(1, 0)$ | $(0, 1)$ | $(1, 1)$ |
|---|---|---|---|---|
| $\varepsilon_{++}$ | 1 | 1 | 1 | 1 |
| $\varepsilon_{+-}$ | 1 | 1 | $-1$ | $-1$ |
| $\varepsilon_{-+}$ | 1 | $-1$ | 1 | $-1$ |
| $\varepsilon_{--}$ | 1 | $-1$ | $-1$ | 1 |

The appearance of the spaces $U^\varepsilon$ will depend on $\widehat{\rho}$. They are however simple to compute—given a vector $v \in V$ and an $\varepsilon \in (\mathbb{Z}_2^2)^*$, the element

$$v_\varepsilon = \sum_{h \in \mathbb{Z}_2^2} \varepsilon(h)^{-1} \widehat{\rho}(h) v$$

is in $U^{\varepsilon}$ [2]

$$\widehat{\rho(k)} v_\varepsilon = \sum_{h \in \mathbb{Z}_2^2} \varepsilon(h)^{-1} \widehat{\rho}(kh) v = \lceil kh = \ell \rceil = \sum_{\ell \in \mathbb{Z}_2^2} \varepsilon(k^{-1}\ell)^{-1} \widehat{\rho}(\ell) v = \varepsilon(k) v_\varepsilon.$$

Consequently, given any basis of $V$, we may construct a spanning set of $U^\varepsilon$ via calculating the above vector for each basis vector, and subsequently extract a basis. Applying this strategy to the example of the Vierergroup acting through flipping on the spaces of $3 \times 3$ convolutional filters yields the bases depicted in Figure 6.

## B.7 A (not very useful) canonical lifting procedure

A natural question is whether every projective representation $\rho$ can be lifted to a linear one. In fact, this can be done, if one allows the group $H$ to depend on $\rho$.

**Definition B.4.** Let $\rho : G \to \mathrm{PGL}(V)$ be a projective representation. We define

$$H_\rho = \{(g, A) \,|\, A \in \rho(g)\} \subseteq G \times \mathrm{GL}(V).$$

---

[2]This strategy is generally applicable. It will be quite inefficient if the group has many elements, though.

**Proposition B.5.** $H_\rho$ *is a subgroup of* $G \times \mathrm{GL(V)}$*, and a covering group of* $G$*. The covering map is given by*

$$\varphi(g, A) = g.$$

*Proof.* The subgroup property follows from the fact that $\rho$ is a projective representation – if $(g, A)$, $(h, B)$ is in $H_\rho$, it per definition means that $A \in \rho(g)$ and $B \in \rho(h)$. However, then

$$AB \in \rho(g)\rho(h) = \rho(gh),$$

i.e. $(g, A) \cdot (h, B) = (gh, AB) \in H_\rho$.

That $\varphi$ is continuous, surjective and maps a neighbourhood of the identity $(e, \mathrm{id}) \in H_\rho$ homeomorphically to a neighbourhood of $e$ in $G$ is clear. Likewise clear is that $\varphi$ is a group homomorphism:

$$\varphi((g, A) \cdot (h, B)) = \varphi((gh, AB)) = gh = \varphi(g, A)\varphi(g, B).$$

$\square$

We can now lift $\rho$ to a representation $\widehat{\rho}$ on $H_\rho$ as follows:

$$\widehat{\rho} : H_\rho \to \mathrm{GL}(V), (g, A) \to A.$$

Indeed,

$$\widehat{\rho}(g, A) = A \in \rho(g) = \rho(\varphi(g, A)),$$

which is the same as saying that $\rho \circ \varphi = \Pi_{\mathrm{GL}(V)} \circ \widehat{\rho}$

It is now however important to note that although this construction is theoretically pleasing, using it in conjunction with Proposition 2.15 does not give much additional insight. Said proposition states that $v$ solves $(\mathrm{Proj}_G)$ if and only if it is the equivalence class of some $x \in U^\varepsilon$ for some $\varepsilon \in H_\rho^*$. However, $x$ being in $U^\varepsilon$ means that

$$Ax = \widehat{\rho}((g, A))x = \epsilon(g, A)x \tag{11}$$

for every $g \in G$ and $A \in \rho(g)$ – that is, for every $A$ in the equivalence class of $g$, $Ax$ is equal to $x$ times some constant that may depend on both $A$ and $g$, which is just a reformulation of $(\mathrm{Proj}_G)$.

For more information about this canonical lift we refer to the textbook (Kirillov, 2012) and the historical exposition (Hirai et al., 2013). The cases where it becomes useful is when $H_\rho$ is a known group with known representations and in particular when $H_\rho = H$ is independent of $\rho$. This is the case for instance for the lift of representations of $\mathrm{SO}(n)$ to $\mathrm{SU}(n)$.

### B.8 Group theoretic facts

Here we, out of convenience for the reader, present proofs of some well-known group theoretical facts.

**Lemma B.6.** *Let* $H$ *be a compact group and* $\widehat{\rho}$ *a representation of* $H$ *on a finite-dimensional space* $V$. *Then there exist an scalar product on* $V$ *with respect to which all* $\widehat{\rho}(h)$, $h \in H$, *are unitary.*

*Proof.* Since $H$ is compact, there exists a unique normalized Haar measure $\mu$ on $H$. That is, $\mu$ is a measure with the property that $\mu(h^{-1}A) = \mu(A)$ for all $h \in H$ and Borel measureable $A \subseteq H$. Given any scalar product $\langle \cdot, \cdot \rangle$ on $V$, we may now define

$$\langle v, w \rangle_H = \int_H \langle \widehat{\rho}(h)v, \widehat{\rho}(h)w \rangle \mathrm{d}\mu(h).$$

This is a well-defined expression due to $V$ being finite-dimensional and $H$ compact (and hence, $h \mapsto \langle \widehat{\rho}(h)v, \widehat{\rho}(h)w \rangle$ is a continuous, bounded function). It is easy to show that it is sesqui-linear. To show that it

is definite, note that if $0 = \langle v, v \rangle_H$, $\langle \widehat{\rho}(h)v, \widehat{\rho}(h)v \rangle$ is a continuous function in $h$ zero almost everywhere, i.e., zero. This means that $\langle \widehat{\rho}(h)v, \widehat{\rho}(h)v \rangle = 0$ for all $h$, which immediately implies that $v$ is zero.

We now claim that any $\widehat{\rho}(k)$ is unitary with respect to this inner product. We have

$$\langle \widehat{\rho}(k)v, \widehat{\rho}(k)w \rangle_H = \int_H \langle \widehat{\rho}(h)\widehat{\rho}(k)v, \widehat{\rho}(h)\widehat{\rho}(k)w \rangle \mathrm{d}\mu(h) = \int_H \langle \widehat{\rho}(hk)v, \widehat{\rho}(hk)w \rangle \mathrm{d}\mu(h)$$
$$= \lceil h' = hk \rceil \int_H \langle \widehat{\rho}(h')v, \widehat{\rho}(h')w \rangle \mathrm{d}\mu(h') = \langle v, w \rangle_H,$$

where we used the invariance of the Haar measure in the substitution step. $\square$

**Lemma B.7.** $S_n^* = \{1, \mathrm{sgn}\}$.

*Proof.* Let $\varepsilon \in S_n^*$ be arbitrary. Since $S_n$ is generated by transpositions $\tau = (ij)$, $\varepsilon$ is uniquely determined by its values of them. Since $\tau^2 = 1$, we must have $\varepsilon(\tau)^2 = 1$, meaning that each value $\varepsilon(\tau)$ is either $+1$ or $-1$. If we can prove that they must all have the same value, we are done: if $\varepsilon$ maps the transpositions to $+1$, $\varepsilon$ is the character 1, and if it maps them to $-1$, it is sgn.

So let $\tau$ and $\tau'$ be to transpositions. We distinguish two cases.

Case 1: $\tau$ and $\tau'$ share an element, i.e., $\tau = (ij)$ and $\tau' = (ik)$. Since $(ik) = (ij)(jk)(ij)$, we must however have $\varepsilon(\tau) = \varepsilon(\tau') \cdot \varepsilon(ij)^2 = \varepsilon(\tau')$.

Case 2: $\tau$ and $\tau'$ do not share an element, i.e., $\tau = (ij)$ and $\tau' = (k\ell)$. Since $(k\ell) = (ik)(j\ell)(ij)(ik)(j\ell)$, we have $\varepsilon(\tau') = \varepsilon(\tau)\varepsilon(ik)^2\varepsilon(j\ell)^2 = \varepsilon(\tau)$. $\square$

**Lemma B.8.** $\{S_n, S_n\} = A_n$.

*Proof.* This proof can be found in e.g. (Ore, 1951).

That $\{S_n, S_n\} \subseteq A_n$ follows from the fact that all commutators have signature 1:

$$\sigma(\tau \circ \pi \circ \tau^{-1} \circ \pi^{-1}) = \sigma(\tau)\sigma(\pi)\sigma(\tau)^{-1}\sigma(\pi^{-1}) = 1.$$

To prove the converse, let us first note that the cases $n = 1, 2$ are trivial—in both those cases, $A_n = \{\mathrm{id}\}$, but $S_n$ is also abelian, so that $\{S_n, S_n\} = \{\mathrm{id}\}$. For $n \geq 3$, we use the well-known fact that $A_n$ is generated by 3-cycles (Lee, 2018, Cor. 6.3) $(ijk)$. If we show that they are commutators, we are hence done. But

$$(ijk) = (jk)(ij)(jk)(ij),$$

so that the claim follows. $\square$

**Lemma B.9.** $\mathrm{SU}(2)$ *is perfect.*

*Proof.* For this proof, we are going to use that elements in $\mathrm{SU}(2)$ can be thought of as unit quaternions $q = (\alpha, v)$, where $\alpha \in \mathbb{R}$ is the real part of the quaternion, and $v \in \mathbb{R}^3$ its vector part. $q$ being of unit norm means that $\alpha^2 + |v|^2 = 1$ The group action is given by the quaternion multiplication, i.e.,

$$(\alpha, v) \cdot (\beta, w) = (\alpha\beta - v \cdot w, \alpha v + \beta w + v \times w),$$

where $\cdot$ and $\times$ are the standard dot and cross products of $\mathbb{R}^3$. We need to show that all unit quaternions can be generated by commutators. We will do this in three steps.

Step 1. Each unit norm quaternion $q$ has a square root $r$, i.e., an $r$ with $r^2 = q$. This can be seen through direct calculation: Each unit norm quaternion can be written as $(\cos(2\theta), \sin(2\theta)u)$ for a $\theta \in [0, \pi]$ and a unit norm $u$. If we then define $r = (\cos(\theta), \sin(\theta)u)$, we have

$$r^2 = (\cos^2(\theta) - \sin^2(\theta), \cos(\theta)\sin(\theta)u + \cos(\theta)\sin(\theta)u + \sin^2(\theta)u \times u)$$
$$= (\cos(2\theta), \sin(2\theta)u) = q.$$

Step 2. Write $r = (\alpha, \beta u)$, let $v$ be a unit norm ector orthogonal to $u$ and $w = u \times v$. If we define $s = (0, v)$, we have

$$sr^{-1}s^{-1} = (0, v)(\alpha, -\beta u)(0, -v) = (0 + \beta v \cdot u, \alpha v - \beta v \times u)(0, -v) = (0, \alpha v + \beta w)(0, -v)$$
$$= (\alpha |v|^2 - \beta w \cdot v, -\alpha v \times v - \beta w \times v) = (\alpha, \beta u) = r,$$

i.e., $r = sr^{-1}s^{-1}$

Step 3. We now combine the above steps

$$q = r^2 = rsr^{-1}s^{-1} = \{r, s\}.$$

Hence, $q$ is a commutator, and the claim has been proven. $\qquad\square$

**Corollary B.10.** $\mathrm{SO}(3)$ *is perfect.*

*Proof.* The perfectness follows from the perfectness of its cover $SU(2)$. If $\varphi : \mathrm{SU}(2) \to \mathrm{SO}(3)$ is a covering map, write any $R \in \mathrm{SO}(3)$ as $\varphi(r)$ for an $r \in \mathrm{SU}(2)$. We just proved that there exists $s, t \in \mathrm{SU}(2)$ with $r = \{s, t\}$. But this implies

$$R = \varphi(r) = \varphi(\{s, t\}) = \{\varphi(s), \varphi(t)\}.$$

$\qquad\square$

**Lemma B.11.** $\mathbb{Z}_n^*$ *is isomorphic to the set of $n$-th roots of unity.*

*Proof.* $\mathbb{Z}_n$ is an abelian group, generated by the element 1. Consequently, any group homomorphism is uniquely determined by the value $\epsilon(1)$. Since $1 = \epsilon(0) = \epsilon(\underbrace{1 + \cdots + 1}_{n \text{ times}}) = \epsilon(1)^n$, $\epsilon(1)$ must be a root of unity. Hence, all elements of $Z_n^*$ is of the form $\epsilon_\omega(k) = \omega^k$ with $\omega$ a root of unity. Since that surely defines a group homomorphism, $\mathbb{Z}_n^* = \{\varepsilon_\omega, \omega^n = 1\}$. As for the isomorphy, we simply need to remark that $\varepsilon_\omega \varepsilon_{\omega'} = \varepsilon_{\omega\omega'}$ by direct calculation. $\qquad\square$

**Lemma B.12.** *If $G$ is perfect, $G^*$ only contains the trivial character 1.*

*Proof.* Let $k = \{h, \ell\}$ be a commutator and $\varepsilon \in G^*$. Then

$$\varepsilon(k) = \varepsilon(h\ell h^{-1} \ell^{-1}) = \varepsilon(h)\varepsilon(\ell)\varepsilon(h)^{-1}\varepsilon(\ell)^{-1} = 1$$

Since $\varepsilon$ is a group homomorphism, this shows that $\varepsilon$ is equal to 1 on the entire commutator subgroup $\{G, G\}$, and thus, since $G$ is perfect, on the entirety of $G$. $\qquad\square$

In fact, to some extent, the converse statement of the last lemma holds. We include the proof for completeness.

**Lemma B.13.** *Let $\mathbb{F} = \mathbb{C}$. If $G$ is a compact group with $G^* = \{1\}$, $G$ is perfect.*

*Proof.* $\{G, G\}$ is a *normal* subgroup of $G$, meaning that if $h \in G$ and $g \in \{G, G\}$, the conjugation $hgh^{-1}$ is also ($\{h, g\} = hgh^{-1}g^{-1}$ is surely in $\{G, G\}$, and therefore also $hgh^{-1} = \{h, g\}g$). This implies that we can define the quotient group $A = G/\{G, G\}$, which is defined via identifying elements $h, k \in G$ for which $hk^{-1} \in \{G, G\}$. The group operation is inherited from $G$: For two equivalence classes $[h]$ and $[k]$, its product is defined as $[h][k] = [hk]$. Also, an $\varepsilon \in G^*$ defines an element of $A^*$. Indeed, any $\varepsilon \in G^*$ must by the proof of Lemma B.12 be identically equal to 1 on $\{G, G\}$. This has the consequence that if $h, k$ are in the same equivalence class,

$$\varepsilon(h) = \varepsilon(hk^{-1}k) = \varepsilon(hk^{-1})\varepsilon(k) = \varepsilon(k),$$

so that $\varepsilon$ is well-defined on $A$.

The group $A$ is called the *Abelianization* of $G$. The naming stems from the fact that it is an abelian group—for any elements $[h]$, $[k]$ of $A$, $\{[h],[k]\} = [\{h,k\}] = [e]$, due to the definition of $A$. Since $G$ is compact, $A$ surely also is.

Since $G^*$ only contains the trivial element, $A^*$ also does. Since $A$ is an abelian compact group, we may now apply the *Pontryagin duality theorem* (Rudin, 1962, p.28), which states that $A$ is isomorphic to the *bidual* $A^{**} = \{1\}^* = \{e\}$. Hence, $A$ only contains the unity element, which in turn means that $k = ke^{-1} \in \{G, G\}$ for every $k \in G$. That is, $G$ is perfect. □

## C  Details of the Spinor Field Networks experiments

As advertised, we begin with a review of spinors and the representation theory of SU(2). The reader is referred to e.g. (Biedenharn & Louck, 1984; Hall, 2013; Zee, 2016) for in depth treatments. SU(2) has one so-called irreducible representation (irrep) for each dimension $n > 0$. Rather than labelling them by $n$, they are typically labelled by $\ell = 0, 1/2, 1, 3/2, \ldots$, where the $\ell$'th irrep (or rather the space it is acting on) has dimension $2\ell + 1$. $\ell = 0$ corresponds to scalar values that don't transform under SU(2) (i.e., SU(2) acts as the identity on them), $\ell = 1$ corresponds to vector values on which SU(2) acts as 3D rotation matrices and $\ell = 1/2$ corresponds to spinor values on which SU(2) acts as the ordinary $2 \times 2$ complex matrix representation of SU(2). For integer $\ell$, the SU(2)-irrep can be taken to consist of only real valued matrices and hence can be interpreted as acting on $\mathbb{R}^{2\ell+1}$. These irreps are also irreps of SO(3). For non-integer $\ell$ on the other hand, it is not possible to let the representation consist of only real valued matrices. Thus these irreps act on $\mathbb{C}^{2\ell+1}$. Furthermore, these irreps do not correspond to linear representations of SO(3), but only to projective representations of SO(3).

Let us denote the irreps of SU(2) by $\widetilde{\rho}_\ell$, $\ell = 0, 1/2, 1, 3/2, \ldots$ and the spaces that they act on by $\mathcal{V}_\ell \sim \mathbb{C}^{2\ell+1}$ (or $\sim \mathbb{R}^{2\ell+1}$ for integer $\ell$). All other finite dimensional representations of SU(2) can be decomposed into these irreps, meaning that if SU(2) acts linearly on $\mathbb{C}^m$, we can decompose the space as

$$\mathbb{C}^m \sim \bigoplus_{\ell \in L} \mathcal{V}_\ell$$

for some specific multiset $L$ of indices such that $\sum_{\ell \in L}(2\ell + 1) = m$. One particular decomposition is the decomposition of the tensor product of two $\mathcal{V}_\ell$'s. Given $\ell \geq \ell'$ we have that

$$\mathcal{V}_\ell \otimes \mathcal{V}_{\ell'} \sim \mathcal{V}_{\ell-\ell'} \oplus \mathcal{V}_{\ell-\ell'+1} \oplus \cdots \oplus \mathcal{V}_{\ell+\ell'-1} \oplus \mathcal{V}_{\ell+\ell'}. \tag{12}$$

In particular, there exists a change of basis matrix that transforms canonical coordinates on $\mathcal{V}_\ell \otimes \mathcal{V}_{\ell'}$ into coordinates which can be split into blocks, each representing coordinates for one $\mathcal{V}_j$. We denote this matrix from the left hand side of (12) to the right hand side by $C$. $C$ consists of so-called Clebsch-Gordan coefficients, which are known and hence it is feasible to perform this decomposition.

The basic idea for our SU(2)-equivariant neural network architecture is now to have layers that combine features and filters using the tensor product. This is a standard approach to constructing SO(3)-equivariant neural networks (Thomas et al., 2018; Geiger & Smidt, 2022; Kondor et al., 2018) which we here generalize slightly to SU(2). Say that we have a feature $f$ that transforms according to $\widetilde{\rho}_\ell$ and a filter $\psi$ that transforms according to $\widetilde{\rho}_{\ell'}$. Their tensor product will transform according to $\widetilde{\rho}_\ell \otimes \widetilde{\rho}_{\ell'}$. Note that (12) means that $C(f \otimes \psi)$ transforms according to

$$C(f \otimes \psi) \xmapsto{\text{SU(2)}} C(\widetilde{\rho}_\ell \otimes \widetilde{\rho}_{\ell'})(f \otimes \psi) = (\widetilde{\rho}_{\ell-\ell'} \oplus \widetilde{\rho}_{\ell-\ell'+1} \oplus \cdots \oplus \widetilde{\rho}_{\ell+\ell'-1} \oplus \widetilde{\rho}_{\ell+\ell'})C(f \otimes \psi).$$

This means that $C(f \otimes \psi)$ is a concatenation of quantities that transform according to known irreps. Since the tensor product is the most general bilinear map, a good way to look at it is that $C$ defines all possible multiplications of $f$ and $\psi$ that are SU(2) equivariant. If we are only interested in a particular output type, say the part of $C(f \otimes \psi)$ that transforms according to $\widetilde{\rho}_{\ell-\ell'}$, then we can simply let $\tilde{C}$ consist of the corresponding rows of $C$, i.e., the first $2(\ell - \ell') + 1$ rows, and the quantity $\tilde{C}(f \otimes \psi)$ will be the output with correct behaviour. $C$ typically contains many zeros and so we don't actually have to compute the complete

$f \otimes \psi$ if we are only interested in some particular output irreps, but can instead only compute those values that will be multiplied by non-zero values of $C$. If our network layers consist of tensor products like this, it is easy to keep track of how the features in the network transform under SU(2), and hence ensure that the output of the network transforms correctly. In our experiments we have outputs that transform according to $\widetilde{\rho}_{1/2}$ and so build networks that guarantee such output.

*Example* C.1. Let us walk through the example of tensoring a vector feature $f$ by a vector filter $\psi$. Both the feature and filter transform according to $\widetilde{\rho}_1$ which we can take to consist of real-valued 3D rotation matrices. This means that according to (12), the output should be decomposable into $\mathcal{V}_0 \oplus \mathcal{V}_1 \oplus \mathcal{V}_2$, i.e., a scalar, a vector and a "higher order" 5-dim. feature. The reader already knows how to combine two vectors to form a scalar and a vector, we use the scalar product and the cross product! Indeed, the tensor product has the following form:

$$(f_1, f_2, f_3)^T \otimes (\psi_1, \psi_2, \psi_3)^T = (f_1\psi_1, f_1\psi_2, f_1\psi_3, f_2\psi_1, f_2\psi_2, f_2\psi_3, f_3\psi_1, f_3\psi_2, f_3\psi_3)^T.$$

The first row of $C$ selects those values that yield the scalar product of $f$ and $\psi$:

$$C_1 = \begin{pmatrix} 1 & 0 & 0 & 0 & 1 & 0 & 0 & 0 & 1 \end{pmatrix}, \quad C_1(f \otimes \psi) = f \cdot \psi.$$

The second to fourth rows of $C$ select those values that yield the cross product of $f$ and $\psi$:

$$C_{2:4} = \begin{pmatrix} 0 & 0 & 0 & 0 & 0 & 1 & 0 & -1 & 0 \\ 0 & 0 & -1 & 0 & 0 & 0 & 1 & 0 & 0 \\ 0 & 1 & 0 & -1 & 0 & 0 & 0 & 0 & 0 \end{pmatrix}, \quad C_{2:4}(f \otimes \psi) = f \times \psi.$$

The last rows of $C$ select values that yield a 5-dim. product of $f$ and $\psi$ transforming according to the irrep $\widetilde{\rho}_2$, we don't write it out here as it is not too illuminating.

### C.1 Tensor spherical harmonics filters

We defined the layers in our Spinor Field Networks like

$$f_i' = \sum_{j \neq i} (\psi(x_j - x_i) \oplus s_j) \otimes f_j. \tag{13}$$

A tensor spherical harmonic (see (Biedenharn & Louck, 1984)) is the tensor product of a spherical harmonic and some other quantity that transforms under SU(2) according to some representation $\widehat{\rho}$. Then (13) could become for instance

$$f_i' = \sum_{j \neq i} (\psi(x_j - x_i) \otimes s_j) \otimes f_j. \tag{14}$$

Technically our filters in (13) are tensor spherical harmonics where the spherical harmonic tensored by $s_j$ is the trivial $Y^0$.

### C.2 Training/implementation details

The implementation is inspired by the `e3nn` package (Geiger & Smidt, 2022), but differs in some crucial aspects (apart from the obvious fact of using spinors).

A tensor product layer in our framework works as follows. We define the number of input, filter and output scalars, spinors and vectors. For each output requested, certain input types can be used. For instance, to produce an output scalar, we can multiply

(i) an input scalar and a filter scalar, or

(ii) an input spinor and a filter spinor, or

(iii) an input vector and a filter vector.

| | Layer 1 | | | | | |
| | Input | | | Filter | | |
| | Scalars | Spinors | Vectors | Scalars | Spinors | Vectors |
| *Spinors as scalars* | 4 | 0 | 1 | 1 | 0 | 1 |
| *Spinors as features* | 0 | 1 | 1 | 1 | 0 | 1 |
| *Spinors as filters* | 0 | 0 | 1 | 1 | 1 | 1 |
| *Spinors squared as vector features* | 0 | 0 | 3 | 1 | 0 | 1 |
| *Spinors squared as vector filters* | 0 | 0 | 1 | 1 | 0 | 3 |
| | Layer 2 | | | | | |
| | Input | | | Filter | | |
| | Scalars | Spinors | Vectors | Scalars | Spinors | Vectors |
| *Spinors as scalars* | 32 | 0 | 8 | 1 | 0 | 1 |
| *Spinors as features* | 32 | 12 | 12 | 1 | 0 | 1 |
| *Spinors as filters* | 32 | 4 | 4 | 1 | 1 | 1 |
| *Spinors squared as vector features* | 32 | 0 | 8 | 1 | 0 | 1 |
| *Spinors squared as vector filters* | 32 | 0 | 8 | 1 | 0 | 3 |
| | Layer 3 | | | | | |
| | Input | | | Filter | | |
| | Scalars | Spinors | Vectors | Scalars | Spinors | Vectors |
| *Spinors as scalars* | 32 | 0 | 8 | 1 | 0 | 1 |
| *Spinors as features* | 0 | 12 | 0 | 1 | 0 | 1 |
| *Spinors as filters* | 32 | 4 | 4 | 1 | 1 | 1 |
| *Spinors squared as vector features* | 32 | 0 | 8 | 0 | 1 | 0 |
| *Spinors squared as vector filters* | 32 | 0 | 8 | 0 | 1 | 0 |

Table 2: Number of scalars, spinors and vectors in each layer of the five network types. The number of outputs of each type is the same as the number of inputs to the next layer. The final layer always outputs one spinor, except for the *Spinors as scalars* net which outputs four scalars.

So for each output scalar requested, we compute a weighted sum of all input scalars and multiply it by a weighted sum of all filter scalars, and similarly for spinors and vectors. These weighted sums contain the learnable parameters in the net. We do not use learnt radial functions as in Tensor Field Networks, as that did not seem to be necessary for our data.

When mapping from spinors to scalars/vectors, we get complex valued scalars and vectors. These are split into real and imaginary parts, hence if 10 output scalars are requested we only compute 5 complex output scalars yielding 10 real output scalars. Note that the operation of taking real and imaginary parts of a vector is rotation equivariant as we work in the basis where rotation matrices are real valued.

The loss used is $L_2$-loss on the regressed spinor, but accounting for arbitrary sign due to projective ambiguity, i.e.

$$\text{loss}(s_{\text{pred}}, s) = \min(\|s_{\text{pred}} - s\|, \|s_{\text{pred}} + s\|).$$

The networks contain between 2500 and 3000 parameters and are small enough to train on CPU in tens of seconds per net. All nets were trained using the Adam optimiser with default PyTorch settings, for 300 epochs with learning rate $10^{-2}$.

The number of features of each type in the layers and filters of the Spinor Field Networks is summarised in Table 2.

## D    Details and further results for ViererNet experiments

We begin by providing some details about the model used in the experiments in the main paper.

The layers of the projectively equivariant ViererNet used in the main paper has, in succession, 32, 32, 32 and 11 intermediate $\mathbb{R}^{n,n}$-valued feature tuples, each of size $4 = (\mathbb{Z}_2^2)^*$. These are subsequently average-pooled to

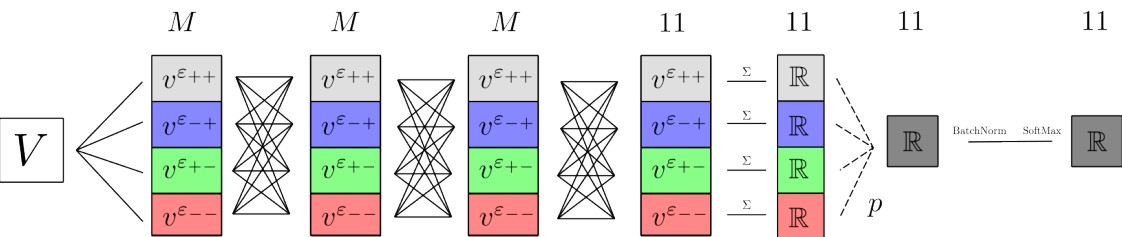

Figure 7: The ViererNet architechtures. $V$ is the input space. For the MNIST experiment, $M = 32$, and for CIFAR, $M = 64$. Best viewed in color.

11 4-tuples. To transform these to scalars, we use 11 selector vectors, and send it through first a batch-norm and then a softmax layer to end up with a final output distribution $p \in \mathbb{R}^{11}$. Before sending the data to the first layer, we normalize it to have zero mean. The base-line model is a three-layer standard CNN, with 32, 32, 32 and 11 $\mathbb{R}^{n,n}$-valued intermediate features, which after average pooling simply feeds their 11 output features through first a batch-norm and then a softmax layer. Both models use tanh non-linearities. The two models have different memory footprints, due to the ViererNet having to handle 4 features in each layer, but each input-output channel pair can be described the same number of parameters (due to Proposition 2.16(ii)). The selector vectors do result in 44 additional ViererNet parameters, but this is miniscule compared to the about 30K in total. The architecture is schematically presented in Figure 7

We train for 100 epochs using the Adam algorithm, with the learning rate parameter set to $10^{-4}$ .

### D.1 Additional experiments on CIFAR10

To complement the MNIST experiments in the paper, we tested ViererNet to classify a similarly modified version of the CIFAR10 dataset.

**Data** We modify the CIFAR10 dataset in two ways. First, we modify the dataset exactly the way MNIST is modified in the main paper: I.e., we add a class 'not an image' which we put (or not, depending on the class) images after they are flipped horizontally or vertically: The images in the classes 'airplane', 'automobile' and 'bird' are never put in the 'not an image'-class, the 'cat' and 'deer' and 'dog' images are put there after a horizontal flip, 'frog' and 'horse' after a vertical flip and 'ship' and 'truck' after both types of flip. We refer to this version of the dataset as the 'both flip' dataset.

The above modification has a problem, and that is that since flipping a CIFAR10-image vertically, in contrast to the MNIST images, most often results in a plausible image. The horizontal flips however results in objects that are upside down, and hence are different from the 'normal' CIFAR images. For this reason, we also consider a different type of modification of the dataset. In this version, we still perform flips with the same probabilities as before, but choose to not change the labels after vertical axis flips at all. We refer to this version of the data set as the 'single flip' datasets.

**Models** We use the same ViererNet and baseline architectures as in the main paper, with the only difference that the number of intermediate features are 64, 64, 64, 11, instead of 32, 32, 32, 11.

**Results** Using the same training algorithm and learning rate (e.g. Adam with $\mathtt{lr} = 10^{-4}$) as for the MNIST experiments, we train each model on each dataset for 100 epochs. The experiments are repeated 30 times. In Figure 8, the evolutions of the median training and test losses are depicted, along with errorbars encapsulating 80% of the experiments. In comparison to the experiments in the main paper, the models are performing more similar. When comparing the models at the epochs with best median performance, the baseline outperforms the ViererNet slightly on both datasets—51.0% vs. 49.9% on the both flip dataset and 59.0% vs. 57.0% on the single flip dataset. The performance difference is however not as significant as in the main paper—the $p$-values for the baseline outperforming ViererNet is only 0.26 for the both flip dataset, and .11 for the single flip dataset.

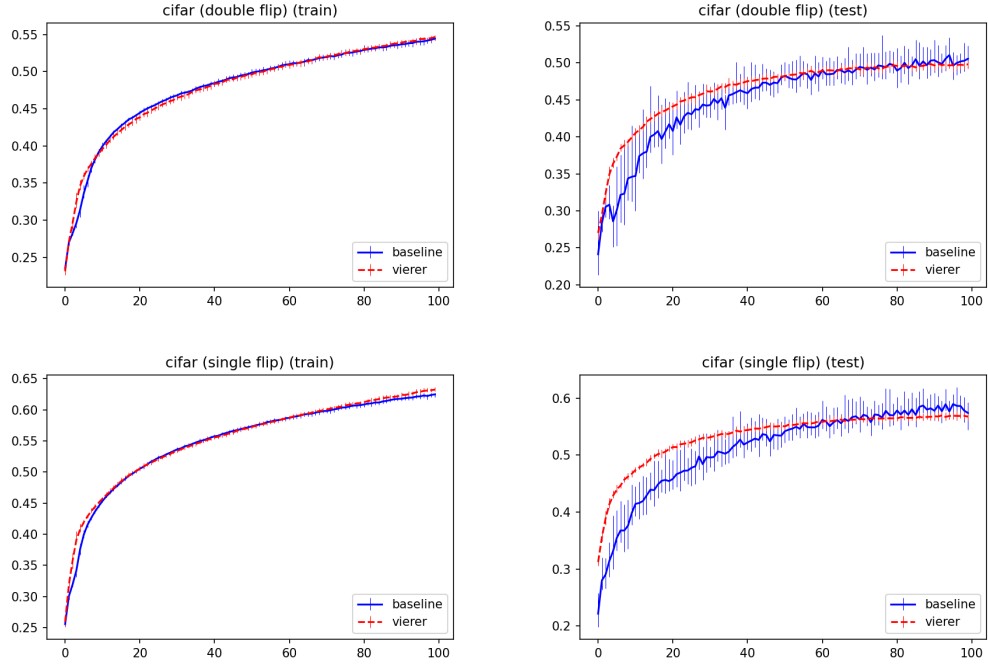

Figure 8: Results for the CIFAR10 datasets. Top left: training accuracy on the 'double flip' version. Top right: test accuracy for the 'double flip' version. Bottom left: training accuracy on the 'single flip' version. Bottom right: test accuracy on the 'single flip' version. The errorbars depict confidence intervals of 80%. Generally, the baseline performs slightly better at the end of the training, but well within the margin of error. The ViererNet is quicker to generalize than the baseline.

| Dataset | MNIST | | | | CIFAR (double flip) | | | | CIFAR10 (single flip) | | | |
| Max it. | 25 | 50 | 75 | 100 | 25 | 50 | 75 | 100 | 25 | 50 | 75 | 100 |
| --- | --- | --- | --- | --- | --- | --- | --- | --- | --- | --- | --- | --- |
| Baseline | 80.3% | 86.6% | 89.4% | 90.2% | 43.2% | 48% | 49.4% | 51.0% | 47.3% | 53.7% | 57.2% | 59.0% |
| ViererNet | **89.6%** | **91.7%** | **92.5%** | **92.8%** | 45.1% | 48.1% | 49.4% | 49.9% | **52.1%** | 55.2% | 56.5% | 57.0% |

Table 3: Median accuracies (over 30 runs) in all experiments for different upper iterations bounds. Statistically significantly ($p < .05$) better performing models for each dataset and upper iteration limits are marked in bold font. (In fact, using the higher $p$-value 0.1 would not change the appearance of the table).

**Performance vs. number of iterations**  Figures 8 and 3 both suggest that the ViererNet can generalize to the test data faster, since it outperforms the baseline in early iterations. To test this quantitatively, we perform the same statistical tests as above, but choose the best performing epoch among the 25, 50 and 75 first iterations instead out of all 100 iterations. We report the median accuracies for all experiments in Table 3. Statistically better figures are highlighted in bold. We see that the ViererNet outperforms the baseline in the MNIST experiments by wider margins, and with high statistical confidence, earlier in the training. Looking at the median values in Table 3, the trend that the ViererNet is quicker to train apparently persists to some extent for the CIFAR experiments. However, a closer looks reveals that we can only with high statistical confidence say that the ViererNet is better than the baseline after 25 iterations on the single flip dataset.

**Discussion**  Although the CIFAR experiments are in general not statistically conclusive, we can conclude that the ViererNet in comparison to the MNIST experiments in the main paper perform worse on the CIFAR set. We believe the main reason behind this is simply that the property of projective $\mathbb{Z}_2^2$-equivariance is less well suited for these tasks. For the both flips version of the dataset, as we discussed above, flipping vertically

will most often lead to a new image which is plausible to come from the same CIFAR-class, and it will be less helpful to apply the strategy of either changing or maintaining the class when flipping the images of the class. As for the single flip version of the dataset, the same argument does not apply. However, in this case, it is ultimately only a subgroup of the Vierergroup that is changing the images in the dataset. Hence, a ViererNet is not the best suited architecture—a $\mathbb{Z}_2$-equivariant would instead be.

A more thorough empirical comparison of the effectiveness of projectively equivariant architectures for classification tasks would be very interesting, but out of scope for this article. We leave it for future work.

### D.2 Training details

We used cross-entropy-loss in our experiments, to account for the different sizes of the classes: the weight vector vectors were chosen as

$$w_{\text{MNIST}} = [1, 1, 1, 1.5, 1.5, 1.5, 1.5, 1.5, 3, 3, 1], w_{\text{CIFAR}} = [1, 1, 1, 1.5, 1.5, 1.5, 1, 1, 1.5, 1.5, 1]$$

for the MNIST and CIFAR10 experiments, respectively. A batch size of 32 was used for all experiments. The models were trained, on single A40 GPUs, in parallell on a high performance computing cluster. The MNIST experiments presented in the main paper used in total about 110 GPU hours, whereas the CIFAR10-experiments presented in the appendix used about 170 GPU hours each.

### D.3 Implementational details

Since all of the basis elements, as depicted in Figure 6 are sparse, we implement the ViererNet layers i PyTorch by saving nine parameters per in-layer-out-layer pair, which we turn into four $3 \times 3$ filters through multiplying the $\mathbb{R}^9$ vector with a sparse matrix. These are then used in padded standard PyTorch convolution layers, and combined in the manner described in Section 3.

