# OpenReview forum: "In search of projectively equivariant networks"
_TMLR — Accepted by TMLR_

### Review · Reviewer_kggx · 2023-10-19

**Summary Of Contributions:**

This paper studies the conditions under which neural network layers are *projectively equivariant*, in the sense that group actions on the input and group actions on the output are equivalent up to multiplication by a nonzero scalar.

The authors describe background of linear and projective equivariance, then prove a series of results about projectively equivariant linear maps. The main result is that projective equivariant linear maps can only lie in certain subspaces, each of which is defined in terms of (the equivalence classes of) linearly equivariant maps scaled by a character of the group. The authors present examples of groups where the only projectively equivariant maps are projections of linearly equivariant ones, and examples of groups where projective equivariance adds additional structure. They further build on this theory by constructing a neural net architecture that is projectively-equivariant by construction.

The authors then conduct a few experiments on synthetic datasets with their proposed architecture. The first is based on MNIST with horizontal and vertical flips, although the motivation of using projective invariance here is pretty weak (see weaknesses below). The second is a toy dataset of point clouds with "spinor features" which jointly transform; for this they find that their proposed architecture outperforms some baselines but not others.

**Audience:**

Yes

**Broader Impact Concerns:**

No broader impact concerns.

**Claims And Evidence:**

Yes

**Requested Changes:**

## Clarity improvements
There were a few parts of the paper I found difficult to follow, and I think the paper could be improved by clarifying them.

- At the end of page 4, you say "For such a relation to hold for generic A, ...". What's a "generic" A in this context? Are you saying essentially "some A are pathological and for those $\varepsilon$ can be something else"? In that case, why aren't those A worth considering?
- In Remark 2.8, I don't understand what you mean by "its image is endowed with $\hat{\rho}^\epsilon$"
- Definition 2.9 and the sentence before it refer to a "so-called covering group of G" and a group covering $\varphi$, but these are never defined except in the Appendix. Please make sure to define these terms or at least point to a definition before defining other terms that depend on them. (In particular, you never say that $\varphi$ maps $H$ to $G$, that it's a homomorphism, or that it's surjective, except in the unreferenced appendix, and all of those seem necessary for understanding your later definitions and results).
- I found the notation in equation $\rho(g) A = \rho_1(g) A \rho_0^{-1}(g)$ to be difficult to understand. Am I understanding correctly that, on the left, $\rho(g) A$ means $\rho(g)[A]$, i.e. you're applying the function $\rho(g)$ to the value $A$ (viewed as a vector), whereas on the right, $\rho_1(g) A \rho_0^{-1}(g)$ means $\rho_1(g) \circ A \circ \rho_0^{-1}(g)$, i.e. you're composing these three linear maps viewed as functions? I'd suggest being more explicit about the notation here, or adding a note clarifying that the implicit concatenation means something different in each case.
- In Section 2.2., you never explain how $S_n$ acts on spaces of tensors (e.g. what the "canonical action" is). I assume this is permuting the tensor elements?
- In Section 3, I don't see how the finitely-many-nontrivial-$U^\varepsilon$ property follows from Proposition 2.14. Could you add a bit more explanation of this? (It's also not obvious to me why we can bound the dimensionality of $U_{HH}$ the way you have shown.)

## Questions

Theorem 2.13 is presented in terms of a covering group $H$ of $G$ and a lift $\hat{\rho}$, and before Definition 2.9 you indicate that you're assuming such a $\hat{\rho}$ exists. Does this mean that there may be some $\rho$ that cannot be lifted to linear representations of any covering group? Or does every projective representation admit a lift, so it's always safe to assume you have one?

In Theorem 2.13, is it necessary to say "$v$ is in the equivalence class of some $x \in U^\varepsilon$"? It seems like if $v$ is in the same equivalence class as $x \in U^\varepsilon$, we'd also have $v \in U^\varepsilon$? Unless you are using $v$ to denote the equivalence class itself, but then shouldn't it be "$v$ **is** the equivalence class of some $x \in U^\varepsilon$", not "is in"?

## Optional suggestions

I wonder if the MNIST example could be replaced with a different toy task where the loss function actually respects the projective invariance equivalence classes. Having the loss depend on the exact scalar value seems to defeat the purpose of designing a projectively equivariant architecture.

I was surprised that in the MNIST example you changed the label when flipping the digit 8 vertically and horizontally. Doesn't an 8 look the same when flipped horizontally or vertically? So it's label intuitively *should* be invariant to flips, and it doesn't make much sense to me to change the label to NaN when flipping it. (That might contribute to the relatively low accuracy of all models for this task.)

For Example 2.6., perhaps it would be worth adding a brief note pointing to Appendix C, since you discuss those concepts in more detail there? I also wonder if it would be helpful to show a visualization similar to your "pinhole camera" figure for SU(2), since it seems that this particular example is used in many times in the paper, and I still don't have an intuitive grasp of what the projective representations mean in this space.

In Figure 2, does it make sense to interpret $V_0$ as being $V_0^1$ in a sense? If so, you could move up the white box to line up with the others and make the figure look a bit more consistent.

In Section 3, I think you could give a bit more motivation for what the $|H^*|$ output features correspond to. At first I was confused why you wouldn't just take $v_K^1$ as the output feature, but perhaps the reason is that which output feature you use determines something about the set of functions you can express? Is that right?


## Minor typographical note

Many of your citations seem to use `\citet` when `\citep` would be more appropriate. For instance, in the first sentence, I think those citations should be entirely in parentheses.

**Strengths And Weaknesses:**

## Strengths

**[S1]** For the most part, I found the paper to be well motivated and clearly written. The authors do a good job of motivating each concept they introduce and explaining the key results informally as well as formally.

**[S2]** The theoretical results seem interesting and general. I liked how the authors also explained the consequences for individual example groups as well.

**[S3]** The proposed architecture makes sense and corresponds nicely to the theoretical results.

## Weaknesses

**[W1]** The MNIST example seems badly motivated and doesn't really feel like it has anything to do with projective invariance. For this task, some classes are invariant to flips and others are not, and the authors note that this could be modeled as either flipping a sign of an activation when the class is flipped, or not flipping the sign; both of these behaviors are projectively equivariant because they differ by only a scalar. Thus, the authors consider using their projectively-equivariant architecture to model it. But assuming I'm understanding correctly, this seems pretty silly, because *every* nonzero scalar function is projectively equivariant; there's only one equivalence class of projectively equivariant scalar-output functions. So if all you need is projective equivariance, any nonzero function would work. And ultimately, the authors end up using a loss which is not well-behaved over projective equivalence classes: it gives different scores to outputs that differ by rescaling, so in a sense it feels like projective invariance is not the right equivalence relation for this task.

I guess this result says something about the inductive biases of the proposed network, but I feel like it doesn't really serve as motivation for projective equivariance, and is sort of trivial from a projective equivariance perspective. If the goal is to handle class-dependent symmetries it seems like it would be easier to directly infer different invariances for each class using something like [Benton et al. (2020)](https://proceedings.neurips.cc/paper/2020/file/cc8090c4d2791cdd9cd2cb3c24296190-Paper.pdf)'s approach.

**[W2]** More generally, it's hard for me to think of situations where a task would be well-suited to projectively invariant architectures like the one proposed here. The spinor fields example in section 4 seems a bit more realistic than the MNIST one, but it's a bit contrived as well.

**[W3]** There were a few parts of the paper that I found hard to follow, and a few terms that did not seem properly defined before they were used. (See the requested changes section below.)

---

> ### Author Response · Authors · 2023-11-27
> **Rebuttal**
>
> We thank the reviewer for the thorough and insightful review. We address the suggestions and questions one by one.
>
> ## Clarity improvements
> - The degenerate $A$ is only the 0-matrix. We have clarified this.
> - "its image is endowed with $\widehat\rho^{\varepsilon}$" means that we let "$\widehat\rho^{\varepsilon}$" act on the output space of $A$. We have rewritten this.
> - We have included the definition of a covering group/group covering in the main text.
> - We have clarified the formulation of $\rho$ prior to Lemma 2.14 (previously Lemma 2.12).
> - We have written out the action of $S_n$ on tensors.
> - "Finitely-many-nontrivial-$U^\epsilon$ property": This really boils down to us assuming that the input, intermediate and output spaces are finite-dimensional when constructing the architecture -- this causes $U$ (and therefore $U_{HH}$) being final dimensional, which means that $\dim(U^\varepsilon)$ can only be non-zero for finitely many $\varepsilon$. This was not properly stated before -- we have improved this.
>
> ## Questions
> - Does every projective representation admit a lift?
> This is a very natural question. The answer is technically yes using something called a central extension of $G$: Given a group $G$ and a projective representation $\rho$, let us define  $H$ as the subgroup $\\{(g,A) : A\in \rho(g)\\}\subset G \times \mathrm{GL}(V)$. That is a cover of $G$ -- the covering map is $\varphi(g,A) = g$ and $\widehat{\rho}(g,A)=A$ defines a lift of $\rho$.
> However, using this construction does not help one much in finding the elements solving $\mathrm{Proj}_G$ -- writing down what $\mathrm{Lin}^\varepsilon_H$ means will simply lead one to a reformulation of $\mathrm{Proj}_G$. It is only when the group $H$ onto which $\rho$ is lifted can be identified as a familiar group, such as in the $\mathrm{SO}(3) \leftrightarrow \mathrm{SU}(2)$ case, the results brings us something.
> We have included a remark (and a discussion in Appendix B.7) about this.
>
> - As for Theorem 2.13, the formulation was wrong before: $v$ denotes the equivalence class itself.
>
> ## Optional suggestions
> Regarding the choice of not changing the label when flipping 8s, one really has to take into account that '8's that are written by humans are not perfectly invariant to flips along either axis. A neural network should be able to lear subtle cues to distinguish flipped 8:s from non-flipped 8:s. Of course, there is an approximate symmetry, so we are making the task a bit harder by making this choice, but still not impossible. We made the choice to not take the approximate symmetry of 8:s into account in order to 'ignore' the specific properties of MNIST, and really approach it as a toy experiment. Since our goal is not to construct a well-performing model, we have chosen not to rerun the experiment with an alternative setup.
>
> Elements in $SU(2)$ are maps between $\mathbb C^2$ and $\mathbb C^2$ -- i.e. maps dealing with objects of real dimension $4$. It is hence very non-trivial to visualize them in an equally insightful manner as the camera model!
>
> The suggestion about the figure is good. We have changed it.
>
>
> Regarding using all $\vert H^*\vert$ features: The reviewer has the right understanding.  Note that $v^1_K$ is a linearly equivariant feature (for each $\varepsilon$, $v^\varepsilon_K$ is '$\varepsilon$'-linearly equivariant). Hence, if we only use that, our network will be linearly equivariant, and hence not be able to represent all projectively equivariant features.

---

> > ### Comment · Reviewer_kggx · 2023-11-29
> > **Discussion**
> >
> > Thanks for answering my questions and making the requested changes to the paper. I also appreciate the new discussion of the central extension as a covering group in the appendix.
> >
> > Regarding the optional suggestions, your responses make sense, thanks. I agree it's probably not necessary to re-run the MNIST experiment given that it's a toy setting, and I appreciate the other clarifications. Not required, but I think you could improve the discussion of the $|H^*|$ features in Section 3 by adding your clarification to the main paper as well (i.e. briefly mention that taking $v^1_K$ just results in a linearly equivariant network and thus doesn't achieve your goal).

---

> > > ### Author Response · Authors · 2023-11-29
> > > **Additional change**
> > >
> > > We are happy that you think our changes make sense! As for the questions $\vert H^*\vert$, we simply missed the possibility to add the clarification to the paper -- we agree that this will benefit the paper, and it is of course no problem to add it. We have uploaded a new version with an additional brief comment in section 3.

---

### Review · Reviewer_2rmv · 2023-10-29

**Summary Of Contributions:**

This paper presents the concept of projective equivariance to the geometric deep learning community, providing a thorough exploration of several crucial questions related to this topic:
- How do projectively equivariant linear layers relate to their linearly equivariant counterparts, and under what conditions do these two types of layers coincide?
- What are the necessary and sufficient conditions for a linear map to exhibit projective equivariance?
- Can we establish a general framework for constructing projectively equivariant neural networks?
- Are there simple examples that empirically validate the superiority of our methods over standard equivariant models?

**Audience:**

Yes

**Broader Impact Concerns:**

I have no concern over the ethical implications of the work.

**Claims And Evidence:**

Yes

**Requested Changes:**

- While demonstrating through experiments is unnecessary for this theoretical work, can the authors provide non-toy applications where considering projective equivariance holds great potential and standard equivariance can get us into trouble? I think the class-dependent symmetry example is already very interesting, but can projective equivariance fully tackle the problem of partial equivariance [1][2]. (It would be super exciting if it can).
- The current writing is not accessible enough to the machine-learning and application audience. I suggest (1) adding high-level intuition of projective equivariance in the introduction. (2) Even though a notation has been introduced before, do add some repetitions in the text to help reading.

[1] Miao, Ning et al. “Learning Instance-Specific Augmentations by Capturing Local Invariances.” International Conference on Machine Learning (2022).
[2] Romero, D. W., & Lohit, S. (2022). Learning partial equivariances from data. Advances in Neural Information Processing Systems, 35, 36466-36478.

**Strengths And Weaknesses:**

**Strengths**
- This paper is technically solid, with all major theoretical results being substantiated by proofs and well-defined conditions. The authors have conducted a comprehensive exploration of all crucial aspects of projective equivariance in deep learning.
- The experiments give simple yet interesting examples, showcasing scenarios where projective equivariance outperforms standard equivariant networks.

**Weaknesses**
- The paper's presentation could be more accessible; readers are required to grasp and remember numerous concepts and notations throughout the text. More repetitions in the context could help. To enhance the paper's readability and appeal, the authors could consider providing high-level intuition in the beginning, followed by the rigorous mathematical statements later on.

---

> ### Author Response · Authors · 2023-11-27
> **Rebuttal**
>
> Thank you for the review. We appreciate the comments regarding readability. We understand the concern that the paper uses many notions which are not standard in the machine learning community, but constantly repeating the meaning of definitions will make the text cluttered, and in particular surpass the page limit. Still, to make the text more accessible, we have expanded the appendix A to include not only the definitions left out in the main text, but all definitions we use. In this way, there is a specific page to resort to if one is unsure what the definition of an object is.
>
> We, unfortunately, do not think that projective equivariance is a simple route to solve the problem of partial equivariance. Partial equivariance is rather equivariance to only certain 'parts' of the group, where as projective equivariance is equivariance to the entire group, but in a different manner than before. It may be that our idea can be transformed to help also in this case, but it will be non-trivial to do so.

---

### Review · Reviewer_SXuJ · 2023-11-13

**Summary Of Contributions:**

In this paper, the authors study projective equivariance in the context of neural networks. The authors present some theory regarding the relation between projectively and linearly equivariant representations, culminating in Theorem 2.13 and the results of Section 2.2. They also propose a way to construct projectively equivariant neural networks, which amounts to building standard equivariant neural networks where the linear group representations acting on each intermediate feature space are lifts of projective group representations. Projective equivariant neural networks are then showcased first for a toy MNIST task and further for a toy spinor regression task where point clouds were equipped with spinor features and the goal was to regress to the spinor targets.

**Audience:**

Yes

**Broader Impact Concerns:**

The paper is of a fairly theoretical nature; as such, I have no ethical concerns and do not believe the paper requires a broader impact statement.

**Claims And Evidence:**

Yes

**Requested Changes:**

This is not crucial for acceptance, but I would like to know if the authors can add some more discussion with respect to my concerns in the weaknesses section above. Notably, I would like to know if they can either introduce or further discuss a better toy motivating example or potentially a real-world application (e.g. from physics) of the introduced networks. Also, a discussion of how the theory may potentially be extended to encompass the spinor example from Section 4 would help tie the paper together.

In terms of writing, I would like to request some changes that would strengthen the presentation of the work. First, I note that the paper uses in-text citations in place of parenthetical citations where they are needed. For example in the first paragraph we see "...to computer vision Krizhevsky et al. (2012)" instead of "...to computer vision (Krizhevsky et al., 2012)"; this should be fixed. Many citation packages have a "\citep" option for parenthetical citations that contrasts with "\citet" for in-text citations. I would like the authors to go through and perform the needed citations corrections. Additionally, I would like to suggest some minor writing corrections that will improve the writing via the following non-exhaustive list:

- In Example 2.1: The authors write "We illustrate in Figure 2." This should say "We illustrate in Figure 1."
- In the paragraph after Example 2.2: "The benefit of projective equivariance is imminent" should be "The benefit of projective equivariance is eminent"
- Stylistic comment about how definitions are given: When first defining "equivariance" and "projective equivariance" it is good to underscore the group action. You can make your definitions more clear by changing e.g. "We refer to $\Phi$ satisfying (4) as being _projectively equivariant_" to "We refer to $\Phi$ satisfying (4) as being _projectively equivariant_ to action by G".
- Stylistic comment: "unit root" on pages 6-7 refers to roots of unity, which is not standard phraseology. I would refer to them simply as roots of unity, e.g., change phrases like "every unit root" to "every root of unity." ("unit root" typically has a different connotation in probability theory/statistics)
- First paragraph of Section 3: "Still, we can still use" -> "Still, we can use"
- Third-to-last paragraph of Section 3: "procedure described in this Section" -> "procedure described in this section"
- Section 4 "Models": "Indata are point clouds" -> "Our data are point clouds"
- Conclusion: "we theoretically studied of the relation" -> "we theoretically studied the relation"

**Strengths And Weaknesses:**

## Strengths

1. Some theory is given to relate projectively equivariant representation with linearly equivariant representations and this characterization later proves useful for the construction of toy projectively equivariant neural networks for a synthetic symmetry-oriented MNIST task.

2. The projective equivariance-oriented approach seems to clearly help neural networks better capture symmetry in the synthetic spinor regression example of Section 4.

## Weaknesses

1. Although the paper does study projective equivariance, which has not yet been explicitly explored in the equivariance literature, the motivating examples do not provide a particularly convincing case for the study of this notion. Namely, the synthetic class-dependent symmetry example with MNIST was very much a toy construction and even then, the model did not outperform a CNN baseline by a very significant amount. Moreover, the paper itself admits that on a synthetic symmetry-oriented CIFAR-10 test task the CNN baseline performs better than the introduced method. Either a more convincing synthetic dataset task or ideally a real-world dataset task (perhaps from physics) as a motivating example for the study of projective equivariance would help bolster the paper significantly.

2. The theoretical results are fairly straightforwardly obtained as is and are not as pertinent as desirable. Notably, they do not hold for the very spinor example given in the paper (Section 4), since the filters and features lie in an infinite dimensional vector space. If not an additional theoretical result, I would have liked to see further discussion of how one might obtain an extension of the extant theory to encompass this highly pertinent case. Additionally, I feel that this paper unfairly differs the design of projectively but not linearly equivariant nonlinearities to future work; such nonlinearities would be highly relevant for constructing more interesting canonical projectively equivariant neural networks and I would have liked to see some attempt at addressing this topic in this paper.

## Verdict

The paper presents an investigation of projective equivariance in the context of equivariant neural networks. The authors present some theory regarding the relation between projectively and linearly equivariant representations and propose a way to construct projectively equivariant neural networks (motivated by the theory). Empirically, projective equivariant neural networks are showcased first for a toy symmetry-oriented MNIST task and further for a toy spinor regression task. Although the motivating examples given are highly synthetic and the theory does not encompass the spinor regression example, I believe the claims made in the submission are well supported and the findings of the paper would be interesting to some portion of TMLR's audience. As such, both criteria for acceptance to TMLR are satisfied; ergo, I recommend an accept rating for the paper.

---

> ### Author Response · Authors · 2023-11-27
> **Rebuttal**
>
> We thank the reviewer for the careful reading and the resulting smaller suggestions for improvement -- they are all reasonable, and we have changed the text accordingly.
>
> As for the more general critique, we first agree that one can always do more experiments to validate and motivate the theory.
> However, the main contribution of the work is an impossibility-result, i.e. that we don't get new types of layers if we use homogeneous coordinates of the projective plane for computer vision application compared to using ordinary cartesian coordinates.
> This is of course hard to validate empirically.
>
> As for the extension to infinite dimensions, we believe our result is already the most crucial step. For a compact group, we can always decompose input- and output space into irreducible, finitedimensional representations, and then apply our theory for each pair of input- and output-irrep.
> Note that what we do in section 4 can be seen as using a reasonable ansatz for the filter-function rather than handling an expansion of it in spaces of functions (corresponding to irreps of SO(3)). The latter involves heavy numerical computations (since we will need to discretize $\mathbb{R}^3$), so we do not think that this approach will bring much. We have adjusted the wording of some passages in section 4 to accompany this discussion.
>
> As for the design of projectively equivariant nonlinearities, we understand the critisism. However, we believe that it will be hard to treat the problem in an as general and systematic way as here. Indeed, characterizing equivariant nonlinearities means to describe *all* equivariant nonlinear functions in a non-trivial way, which will be more or less impossible without additional assumptions on V,W and G (or indeed the nonlinear functions). It will be much more viable to attack the problem on a case by case basis, which is really different from what we do in this paper.

---

> > ### Comment · Reviewer_SXuJ · 2023-12-10
> > **Discussion**
> >
> > Thanks for making the requested changes! I have noted your comments and have made my final recommendation.

---

### Decision · Action_Editor_NNZa · 2023-12-18

**Recommendation:** Accept as is

**Comment:**

All reviewers unanimously agreed that the claims of the submission are well supported, and that the paper can be of interest to part of the TMLR audience, since projective equivariance has not been discussed much in the literature. Therefore, the bar for acceptance at TMLR is met, and the decision is to accept this paper.

**Audience:**

To my knowledge and that of the reviewers, projective equivariance has not been discussed much in the existing literature yet. Therefore, despite the modest evaluation on only toy examples, the paper can still be of interest to part of the TMLR audience and inspire future research in this direction.

**Claims And Evidence:**

All reviewers agreed that the claims made in the submission are well-supported.